



# Tropical cirrus clouds of convective and non-convective origins

Qin Huang[1] and Tra Dinh[1]

[1]Department of Physics, University of Auckland, Auckland, New Zealand

**Correspondence:** Tra Dinh (t.dinh@auckland.ac.nz)

**Abstract.** The occurrence of cirrus clouds in the tropics ($24\,°$S–$24\,°$N) is analyzed using the 2007–2015 monthly data from the Cloud-Aerosol Lidar and Infrared Pathfinder Satellite Observation (CALIPSO) and the fifth generation reanalysis product (ERA5) of the European Centre for Medium-Range Weather Forecasts. In most cirrus clouds, the specific humidity (SPH) is larger than in cloud-free air and/or the temperature is smaller than in cloud-free air. Both positive SPH perturbations and

negative temperature perturbations increase the relative humidity, resulting in favorable conditions for the formation and maintenance of clouds. The clouds in which there are positive SPH perturbations are considered to originate from convection. This is because, in the free troposphere, positive SPH anomalies are largely produced by the upward transport of moisture by convection followed by detrainment of the convective plumes. The remaining clouds that are not directly influenced by convection are driven by negative temperature perturbations. These temperature-driven clouds are formed and maintained in

the cold phases of gravity waves and/or by the adiabatic cooling associated with the upwelling branch of the Brewer–Dobson circulation. Averaged over all altitudes of the tropical atmosphere, there are about three times more convective cirrus than non-convective ones. The level of maximum convective cirrus occurrence is at $14\,$km, i.e., the bottom of the tropical tropopause layer (TTL). Non-convective cirrus obtain their maximum frequency of occurrence at about $16\,$km, which is below the cold point tropopause (CPT). The seasonal cycle of convective cirrus is consistent with that of tropical convection, while the sea-

sonal cycle of non-convective cirrus in the TTL is consistent with that of the CPT. There are two maxima in the frequency of occurrence of convective cirrus, one at around $10\,°$S in the austral summer, and the other at around $10\,°$N in the boreal summer. In contrast, non-convective cirrus occur most frequently near the equator in the boreal winter. The ice water content (IWC) in both convective and non-convective cirrus increases with increasing temperature (decreasing altitude). Thus, non-convective cirrus—which on average occur at lower temperatures (higher altitudes)—tend to have lower IWCs than convective cirrus.

## 1    Introduction

Cirrus are ice clouds which are typically found in the cold atmosphere above $6\,$km–$8\,$km. Cirrus clouds occur as frequently as $20\,\%$ to $70\,\%$ of the time over the different regions of the globe (Wang et al., 1996; Mace et al., 2009; Hong and Liu, 2015; Heymsfield et al., 2017). Their radiative effects significantly influence the dynamics and thermodynamics of the atmosphere (Liou, 1986). To date, the roles of the different processes that govern the occurrence of cirrus clouds remain not well quantified.

This contributes to uncertainties in estimating cirrus cloud amount and their spatiotemporal distribution and radiative effects in models (see e.g. Boucher et al., 2013).





The purpose of this paper is to quantify the roles of convection and non-convective processes in governing the occurrence of cirrus clouds. We focus on the tropics only, given that cirrus clouds are widespread in the tropics (e.g. Sassen et al., 2008; Heymsfield et al., 2017). Furthermore, cirrus clouds in the tropical tropopause layer (TTL) affects the transport of air (Corti

et al., 2005, 2006; Dinh and Fueglistaler, 2014a) and water vapor (see e.g. Wang et al., 1996; Jensen et al., 2001; Dinh and Fueglistaler, 2014b) into the stratosphere, thereby affecting the concentration of water vapor in the stratosphere. Stratospheric water vapor itself plays a significant role in the Earth's radiative energy budget (Solomon et al., 2010; Dessler et al., 2013).

Cirrus clouds form by either freezing of liquid cloud droplets at temperatures above −38 °C or in situ nucleation of ice crystals from the vapor phase at temperatures below −38 °C (Heymsfield et al., 2017). Convection plays a complicated role

in driving cirrus formation from both the liquid and vapor phases. Cirrus clouds that form by freezing of liquid droplets in mixed-phase clouds can be considered to originate from convection. On the other hand, not all cirrus clouds that form in situ from the vapor phase are driven by convection. In the tropics, the negative temperature anomalies that can drive ice nucleation arise from multiple convective and non-convective sources, including (i) the adiabatic or diabatic cooling at the top of deep convection (Hartmann et al., 2001; Sherwood et al., 2003; Robinson and Sherwood, 2006; Kim et al., 2018; Gasparini et al.,

2019), (ii) the adiabatic cooling associated with the upwelling branch of the Brewer–Dobson circulation (BDC, Holton et al., 1995), (iii) large-scale Kelvin and Rossby waves (Boehm and Verlinde, 2000; Immler et al., 2008; Fujiwara et al., 2009; Virts et al., 2010) and small-scale gravity waves (Garrett et al., 2004; Dinh et al., 2016; Kim et al., 2016; Reinares Martínez et al., 2021), and (iv) midlatitude intrusions (Waugh and Polvani, 2000; Taylor et al., 2011).

The percentage of tropical cirrus clouds that originate from convection has been estimated previously using a variety of

methods, each with its own drawback and associated uncertainty. Wang and Dessler (2012) classified cirrus in the TTL that have ice water contents (IWCs) exceed the ambient water vapor to be of convective origin. However, although cirrus of convective origin may have large IWCs at the beginning of their life cycles, subsequent processes such as cloud horizontal spreading and ice sublimation can decrease the IWC by several orders of magnitude (Dinh et al., 2010, 2012, 2014). Another method is using low values of the outgoing longwave radiation (OLR) as a proxy for deep convection, and cirrus clouds located in regions of

low OLRs can be considered to originate from convection (e.g., Massie et al., 2002; Dessler et al., 2006). However, anvil cirrus may persist after the convection has ceased, or they are blown off away from the convective cores. These clouds originate from convection but may not be classified as so using the local OLR proxy. A more sophisticated method to track cirrus clouds in relation to convection is using parcel trajectories. The trajectories are often initialized at the locations of the cirrus clouds and then calculated backward following the winds for a time period (Pfister et al., 2001; Massie et al., 2002; Spang et al., 2002;

Mace et al., 2006). If convection is encountered along the back trajectories, then the clouds at the initialized locations are assumed to be convectively generated. This assumption may overestimate the number of clouds that originate from convection. Even though the convection occurs before the clouds in the same trajectories, it may or may not be the cause for the occurrence of the clouds. Furthermore, if a trajectory is needed to track every cloud, the calculation becomes computationally expensive for a large number of clouds over a large spatial and temporal domain. Other variations of the trajectory method were carried

out by Luo and Rossow (2004) and Ueyama et al. (2015). Ueyama et al. (2015) calculated backward trajectories that end at the tropopause while Luo and Rossow (2004) performed forward trajectories that start from deep convective events; both groups



simulated the evolution of clouds along the trajectories. The relationship between convection and clouds along the trajectories can be analyzed but the results are applicable to the simulated cloud population, rather than the observed cloud population.

Here, we propose a different method to identify cirrus clouds of convective and non-convective origins. It hinges on the physical argument that convection results in a net upward transport of water vapor (Sherwood et al., 2010). Positive specific humidity (SPH) anomalies in the tropical upper troposphere, TTL and lower stratosphere can be in principle traced back to convection. Observational evidence supporting the role of convection in moistening the tropical upper troposphere, TTL and lower stratosphere is available (see e.g. Soden and Fu, 1995; Liao and Rind, 1997; Sassi et al., 2001; Folkins and Martin, 2005; Wright et al., 2009; Corti et al., 2008; Schiller et al., 2009). Accordingly, we identify the cirrus clouds that occur at times when the air contains more moisture than usual to originate from convection. On the other hand, the cirrus clouds that occur when the local conditions are dry and cold are classified to be of non-convective origins. We find that the spatiotemporal distributions of cirrus clouds of convective and non-convective origins are distinct from each other. The seasonal cycle of the cirrus that originate from convection is consistent with that of tropical convection, while the seasonal cycle of the cirrus in the TTL that do not originate from convection is consistent with that of the tropical cold point tropopause (CPT). These results suggest that the new method is indeed appropriate for the purpose of identifying clouds of convective and non-convective origins.

The remaining of the paper is organized as follows. The data and methodology are described respectively in Sections 2 and 3. Section 4 discusses the characteristics of the occurrence of tropical cirrus of convective and non-convective origins, including their spatiotemporal distributions and IWCs. Section 5 contains the summary.

## 2 Data

We analyze the monthly-mean, three-dimensional cirrus cloud occurrence and IWC of the Lidar Level 3 Ice Cloud Data, Standard Version 1-00 (NASA Langley Atmospheric Science Data Center, 2018). This data was collected with the Cloud-Aerosol LIdar with Orthogonal Polarization (CALIOP) instrument on the Cloud-Aerosol Lidar and Infrared Pathfinder Satellite Observation (CALIPSO, Winker et al., 2010). CALIOP is capable of detecting clouds with optical depths of 0.01 or less (Winker et al., 2007). For this reason, CALIOP data are well suited for studies of cirrus clouds, many of which are optically thin. The user guides for the monthly Lidar Level 3 Ice Cloud Data can be found online (see https://www-calipso.larc. nasa.gov/resources/calipso_users_guide/qs/cal_lid_l3_ice_cloud_v1-00.php and https://www-calipso.larc.nasa.gov/resources/ calipso_users_guide/qs/cal_lid_l3_cloud_occurrence_v1-00.php).

The spatial resolution of the monthly Lidar Level 3 Ice Cloud Data is 2.5° in the zonal direction, 2.0° in the meridional direction, and 120 m in the vertical. The monthly data is not suitable to conduct case study of clouds that occur in response to individual convection or wave events. However, if convection and wave activities have intrinsic seasonal cycles, their effects on the seasonal cycles of cirrus occurrence should be captured in the monthly data. Thus, this data is appropriate for us to study the spatial distribution of cirrus clouds on the seasonal and climatological time scales. Figure 1 shows the frequency of occurrence of cirrus clouds in the tropics between 24 °S and 24 °N over the 9-year period from January 2007 to December 2015 obtained





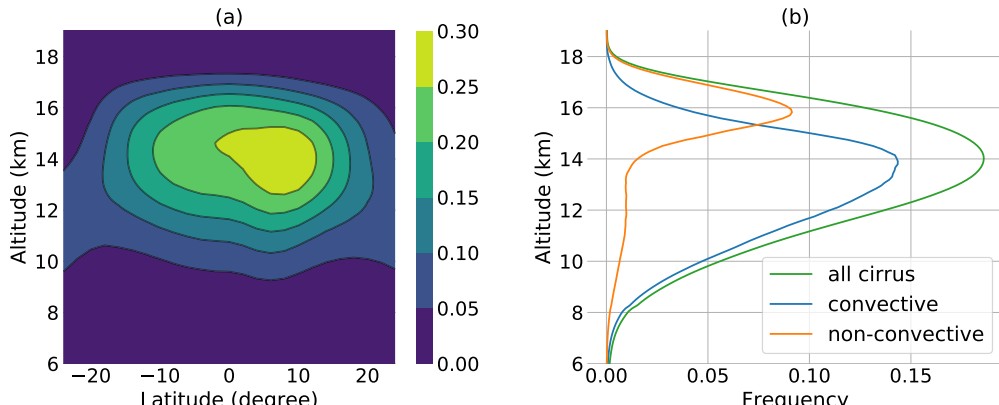

**Figure 1.** Climatological mean frequency of occurrence of cirrus clouds in the tropics: (a) latitude–altitude profile of the zonal mean frequency and (b) vertical profile of the frequency averaged over the tropics.

from CALIPSO. The overall spatial distribution of cirrus clouds in the figure is consistent with previous observational studies
using CALIPSO (Sassen et al., 2008; Mace et al., 2009; Hong and Liu, 2015) and satellite radiometers (Wang et al., 1996).

To study the meteorological conditions surrounding cirrus clouds, we analyze the temperature and SPH of the atmosphere.
For these, we use the fifth generation reanalysis product (ERA5) of the European Centre for Medium-Range Weather Forecasts
(ECMWF). In addition, we obtain the data for precipitation from the Version-2 Global Precipitation Climatology Project
(GPCP) Monthly Precipitation Analysis (Adler et al., 2003). The temperature, SPH, and precipitation data were downloaded
for the same temporal and spatial domains as the ice cloud data above, and then they were interpolated to the same grid as the
ice cloud data.

## 3 Identifying cirrus of convective and non-convective origins

Figure 2 shows the vertical profiles of the relative humidity (RH) that has been averaged over time and the tropical domain
in cloudy and cloud-free conditions. The RH discussed in this work is specifically with respect to ice. At a given grid box
location, the cloudy conditions refer to the times when clouds are detected in the grid box. The cloud-free conditions refer to
the remaining times when the grid box is cloud-free. The figure shows that on average the RH is greater in cloudy conditions
than in cloud-free conditions at every altitude. This is consistent with existing observations (Sandor et al., 2000; Kahn et al.,
2008, 2009; Krämer et al., 2020) that the RH is greater in cloudy conditions to support the formation and maintenance of the
clouds.

The RH is related to the temperature ($T$) and the SPH ($q$) via

$$\text{RH} = \frac{p_{\text{v}}}{e_{\text{si}}(T)} = \frac{R_{\text{v}}}{R} \frac{q}{e_{\text{si}}(T)} p, \tag{1}$$



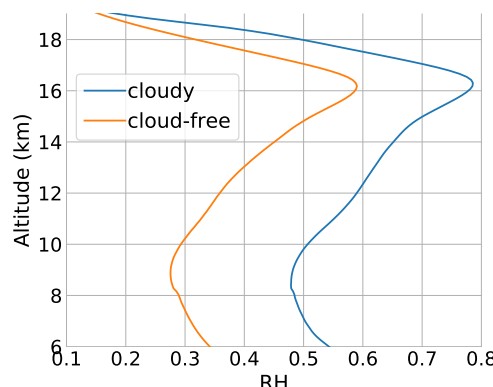

**Figure 2.** Vertical profiles of the climatological mean RH in cloudy and cloud-free conditions averaged over the tropics.

where $R = 287\,\mathrm{J\,kg^{-1}\,K^{-1}}$ and $R_\mathrm{v} = 461\,\mathrm{J\,kg^{-1}\,K^{-1}}$ are respectively the specific gas constants of air and water vapor, $p$ is atmospheric pressure, $p_\mathrm{v}$ is the partial pressure of water vapor in air, and $e_\mathrm{si}(T)$ is the saturation water vapor pressure with respect to ice, a function of temperature. The function $e_\mathrm{si}(T)$ increases with temperature and is calculated based on the empirical

formula given by Murphy and Koop (2005). According to Eq. (1), at a given pressure level, large RH values inside clouds relative to cloud-free conditions must arise from positive SPH anomalies and/or negative temperature anomalies.

Let $\Delta q(x,y,z,t) = q_\mathrm{cld}(x,y,z,t) - \overline{q_\mathrm{cfr}}(x,y,z)$ denote the difference between the SPH in the cloud sample at the location $(x,y,z)$ and time $t$ and the average SPH in cloud-free conditions at the same location. The average cloud-free SPH $(\overline{q_\mathrm{cfr}})$ is obtained by averaging the SPH over the times when that location is cloud-free. Similarly, $\Delta T(x,y,z,t) = T_\mathrm{cld}(x,y,z,t) - \overline{T_\mathrm{cfr}}(x,y,z)$ is the

difference between the temperature in the cloud sample at the location $(x,y,z)$ and time $t$ and the average temperature in cloud-free conditions at the same location. The vertical profiles of the climatological mean, tropical average $\Delta q$ and $\Delta T$ are shown in Fig. 3. The figure shows that $\Delta q$ (green line) is positive in the troposphere, indicating that most cirrus clouds in the troposphere are formed and maintained in the months of positive SPH anomalies. Furthermore, $\Delta q$ decreases exponentially with altitude, consistent with the fact that the background SPH decreases exponentially with altitude. On the other hand, the magnitude of the

temperature anomalies experienced by cirrus clouds is small in most of the troposphere, and it becomes significant only above 14 km or so. The result that cirrus clouds above 14 km experience significant negative temperature anomalies is consistent with previous studies (Boehm and Verlinde, 2000; Virts et al., 2010; Virts and Wallace, 2010; Tseng and Fu, 2017).

In the free troposphere, positive SPH anomalies are largely produced by the upward transport of moisture by convection. Some positive SPH anomalies may be located away from active convection, but even in these cases the source of moisture must

be the convective outflows that have been transported horizontally by the winds (see Salathé and Hartmann, 1997; Sohn et al., 2008; Das et al., 2011). Therefore, we identify the clouds in which $\Delta q > 0$ to be of convective origin, hereafter 'convective' cirrus. Our definition of convective cirrus includes clouds that form within the convective updrafts and at the top of convection, as well as those that form in the moist air of the convective outflows downstream of convection. Examples of the latter type of





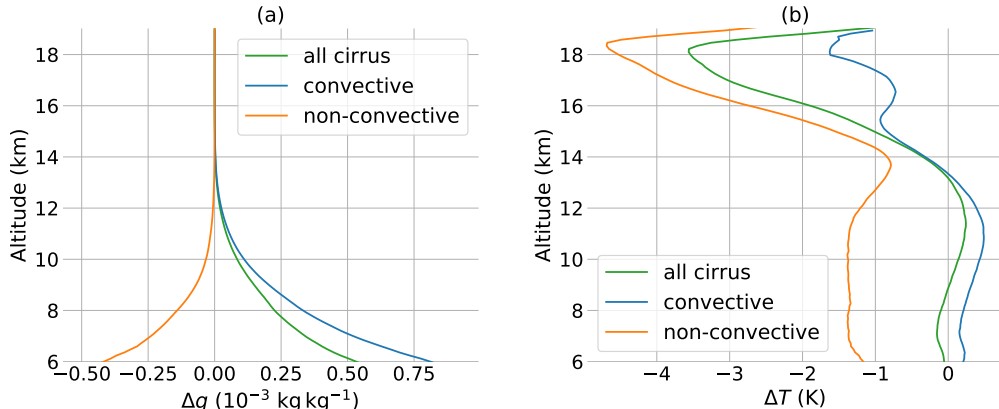

**Figure 3.** Vertical profiles of the climatological mean, tropical average differences in (a) SPH and (b) temperature between cloudy and cloud-free conditions.

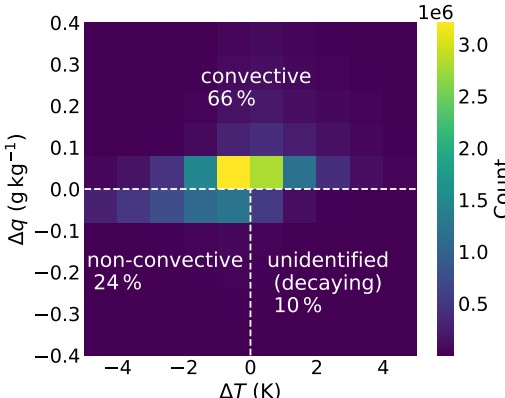

**Figure 4.** Histogram of cloud samples against the differences in SPH and temperature between cloudy and cloud-free conditions.

convective cirrus were recently reported by Cairo et al. (2021). Using this method, we find that 66 % of tropical cirrus clouds

are convective cirrus (see Fig. 4). About half of the convective cirrus experience positive temperature anomalies, and the other

half experience negative temperature anomalies.

The remaining cirrus clouds in which $\Delta q \leq 0$ consist of two categories. The first category comprises of the cloud samples in which $\Delta q \leq 0$ and $\Delta T < 0$. This makes up 24 % of tropical cirrus clouds (see Fig. 4). As these clouds coincide with dry anomalies, the negative temperature anomalies that form and maintain these clouds are unlikely to be the cooling at the top

of convection. Rather, they are associated with waves and/or the adiabatic cooling associated with the upwelling of the BDC.

Even though some waves are generated by convection, the impact of convection on these clouds through wave generation is indirect only. We therefore label these clouds 'non-convective' cirrus. The last category of clouds is for those in which $\Delta q \leq 0$





and $\Delta T \geq 0$. These clouds are neither driven by positive SPH anomalies nor negative temperature anomalies. They are most likely in the decaying stage of their lifetimes. In these cases, the SPH and temperature anomalies cannot be used to identify

their formation and maintenance mechanisms. These clouds are labeled 'unidentified'.

With the unidentified clouds comprising only 10 % of tropical cirrus clouds, the method described above allows us to identify the majority of cirrus clouds and their relationship with convection. Furthermore, as shown in Section 4 below, the spatial distributions and seasonal cycles of convective and non-convective cirrus are distinct from each other. The occurrence of convective cirrus is consistent with the location and the seasonal cycle of tropical convection, while the occurrence of non-

convective cirrus is consistent with the CPT. The seasonal cycle of the CPT is strongly coupled to that of the BDC (Highwood and Hoskins, 1998; Jucker and Gerber, 2017). These results suggest that the method we propose is appropriate to separate convective and non-convective cirrus.

## 4 Characteristics of the occurrence of convective and non-convective cirrus

### 4.1 Spatial distributions

Figures 1 and 5(a) show that the frequency of occurrence of convective cirrus is maximum at around 14 km ($\sim 150$ hPa), coincided with the level of zero net radiative heating rate, which is often defined as the bottom of the TTL (Fueglistaler et al., 2009). The 14 km altitude is also approximately the level of neutral buoyancy, which provides the upper bound for convective development in the vertical (Takahashi and Luo, 2012). The level of maximum convective mass outflow is located several kilometers lower at around 10 km–11 km (Takahashi and Luo, 2012). Convective cirrus between the level of neutral buoyancy

(14 km) and the level of maximum convective outflow (10 km–11 km) are likely anvil cirrus. Convective cirrus above 14 km are likely to originate from (i) the further lofting, spreading and detachment of anvils, (ii) in situ ice nucleation in the moist air of the convective outflows in response to cold anomalies (see Fig. 3) associated with the cooling at the top of deep convection and/or waves. At lower altitudes (below 10 km or at temperatures above 235 K), convective cirrus originate from mixed-phase clouds (Heymsfield et al., 2017), i.e. they are of liquid origin (terminology following Krämer et al., 2016). Convective cirrus

below 14 km tend to experience positive temperature anomalies (see Fig. 3), most likely associated with the latent heat release in convection.

Non-convective cirrus tend to occur at higher altitudes than convective cirrus. The frequency of occurrence of non-convective cirrus maximizes at around 16 km, below the CPT (see Figs. 1 and 5b). The climatological tropical mean CPT is found to be at 16.8 km. The level of maximum cirrus occurrence is capped above by the CPT potentially because of two reasons. Firstly, the

RH decreases with altitude above the CPT as temperature increases with altitude (see Fig. 2). Thus, above the CPT the negative temperature perturbations must be of large magnitudes to raise the RH above the threshold of ice nucleation. Secondly, a necessary condition for cirrus clouds to self-maintain for a long time is that the temperature in the cloud layer decreases with altitude. In this situation, the circulation induced by the cloud radiative heating produces in-cloud water vapor flux convergence that acts against ice sublimation (Dinh et al., 2010). On the other hand, when the temperature in the cloud layer increases with

altitude (such as above the CPT), the circulation induced by the cloud radiative heating produces in-cloud water vapor flux





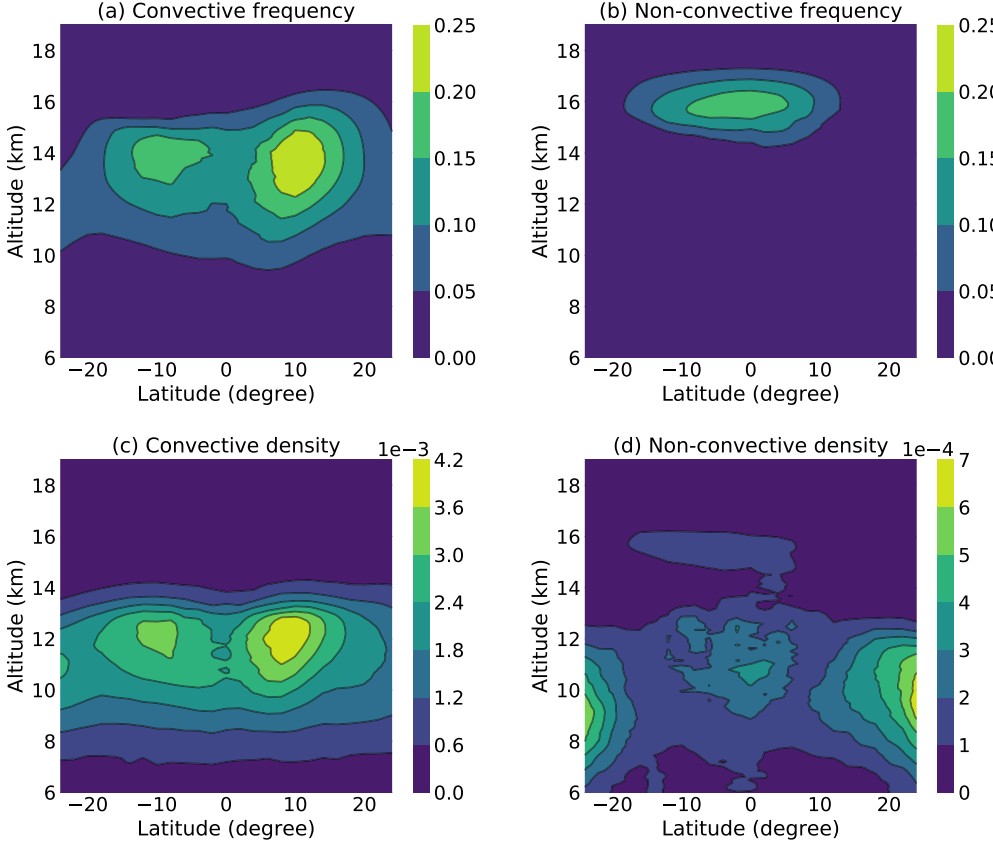

**Figure 5.** Latitude–altitude profiles of the climatological zonal mean frequency of occurrence (top) and grid-average ice mass density in the domain (g m$^{-3}$, bottom) of convective cirrus (left) and non-convective cirrus (right).

divergence that enhances ice sublimation. This means that clouds above the CPT are short-lived and as a result, the frequency of cloud occurrence above the CPT is small.

**Table 1.** Percentage contributions of the different types of cirrus clouds to the total cirrus occurrence in different layers of the tropical atmosphere. The bottom of the tropical tropopause layer (TTL) is located near 14.0 km ($\sim 150$ hPa). The climatological tropical mean CPT is at 16.8 km ($\sim 100$ hPa).

|  | Convective | Non-convective | Unidentified |
|---|---|---|---|
| Above CPT | 23 | 76 | 1 |
| Above 14.0 km | 49 | 44 | 7 |
| All altitudes | 66 | 24 | 10 |





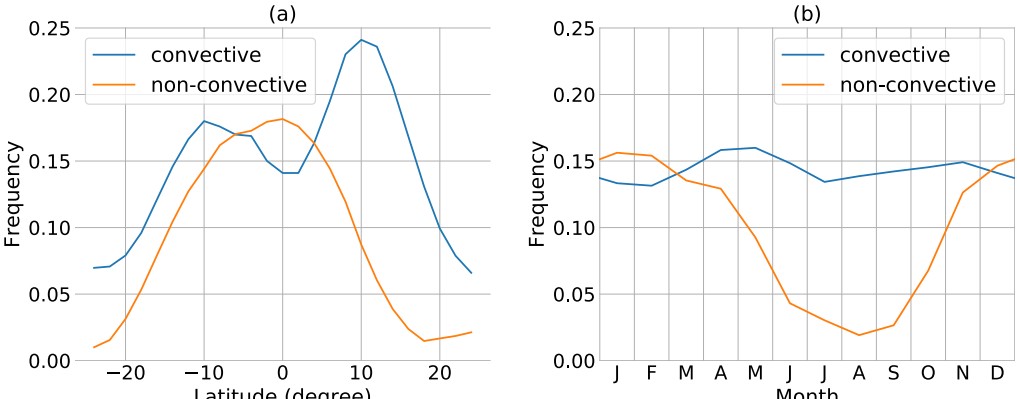

**Figure 6.** Frequency of occurrence of convective and non-convective cirrus clouds: (a) latitudinal profile of the vertical maximum of the climatological zonal mean frequency and (b) climatological monthly profile of the vertical maximum frequency averaged over the tropics.

Table 1 shows the percentage contributions of the different types of cirrus clouds to the total cirrus in different layers of the tropical atmosphere. The table shows that convective cirrus dominate the entire atmosphere and the troposphere below 14 km, i.e., the bottom of the TTL. Above 14 km, convective and non-convective cirrus contribute almost equally to the total cirrus cloud occurrence, a result consistent with Massie et al. (2002) although they studied cirrus clouds over the maritime continent only. Above the CPT, non-convective cirrus make up the large majority (76 %) of clouds, but the percentage of convective cirrus is not negligible (23 %). Note that these results were obtained for the spatially and temporally varying CPT, rather than the climatological tropical mean CPT at 16.8 km.

Based on the vertical profile of the frequency of occurrence of convective cirrus clouds (see Fig. 1b), we can estimate the degree of overshooting convection above the CPT. Above the CPT, the frequency of occurrence of convective cirrus decreases with altitude, indicating that the degree of penetration of convection into the stratosphere decreases with altitude. At the CPT, the frequency of occurrence of convective cirrus is 1.7 %. This provides the upper bound for the occurrence of overshooting convection injecting ice into the stratosphere because not all convective cirrus are formed within the convective updrafts; some convective cirrus are formed in situ in the moist air of the convective ouflows. Gettelman et al. (2002) found based on cloud brightness temperatures that convection is present above the CPT about 0.5 % of the time, which is indeed less than the upper bound estimated here.

The meridional pattern of convective cirrus occurrence is bimodal and asymmetric about the equator (Figs. 5a and 6). There are two maxima at approximately 10 °S and 10 °N, with the northern hemisphere (NH) maximum being larger than the southern hemisphere (SH) maximum, consistent with the fact that convection is stronger in the NH. In comparison, the meridional pattern of non-convective cirrus occurrence is unimodal, with the maximum frequency of occurrence centered around the equator (Figs. 5b and 6). The different spatial distributions of convective and non-convective cirrus suggest that the





mechanisms governing the occurrence of non-convective cirrus is distinct from convection. This topic is further discussed in
Section 4.2.

Figures 5(c) and (d) show the grid-average ice mass density associated with convective and non-convective cirrus. For both
types of clouds, the maximum ice mass density is located below the maximum frequency of occurrence. This is because cirrus
clouds at lower altitudes (higher temperatures) contain more IWCs (more on this in Section 4.3). Interestingly, Fig. 5(d) reveals
that non-convective cirrus in the troposphere contribute significantly to the ice mass in the domain. These low-altitude non-
convective cirrus occur much less frequently than their non-convective counterpart in the TTL (see Fig. 1b). However, they

contain significantly higher IWCs than high-altitude cirrus clouds. Low-altitude non-convective cirrus are located at higher
latitudes towards the northern and southern edges of the tropics, in contrast to high-altitude non-convective cirrus which are
located near the equator (comparing Figs. 5b and d).

### 4.2   Seasonal cycles

Figure 7 shows how the seasonal cycle of convective cirrus is forced from the bottom up by the seasonal cycle of the SPH

in the troposphere, while the seasonal cycle of non-convective cirrus is forced from the top down by the seasonal cycle of
the temperature in the TTL. In each hemisphere, convective cirrus occur most (least) frequently during the summer (winter)
months when the SPH perturbations are positive (negative). The seasonal cycle of convective cirrus in the NH is thus opposite
of that in the SH, i.e., when the frequency of occurrence of convective cirrus is maximum in the NH, it is minimum in the SH.
The net result is that over the entire tropics, the frequency of occurrence of convective cirrus is roughly constant (Fig. 6b). In

contrast, in both the NH and SH non-convective cirrus occur most (least) frequently in the boreal winter (summer) when the
temperatures in the TTL are minimum (maximum). The net result is that over the entire tropics, the frequency of occurrence of
non-convective cirrus has a strong seasonal cycle with a maximum in the boreal winter and a minimum in the boreal summer
(Fig. 6b).

    Figure 8(a) shows the seasonal migrations of convective cirrus and precipitation between the NH in the boreal winter and

the SH in the austral summer. The similar seasonal patterns of convective cirrus and precipitation suggest that these clouds
are indeed coupled to tropical convection. The seasonal variations of convective cirrus are thus controlled by the seasonally
varying Hadley cells, the intertropical convergence zones (ITCZ), and monsoons. The maximum frequency of occurrence of
convective cirrus clouds occur at around $10\,°N$ in the boreal summer and $10\,°N$ in the austral summer, with the boreal summer
maximum larger than the austral summer maximum. The asymmetry between the NH and SH maxima is associated with the

asymmetry of the ITCZ, which arises from the different shapes of the continents in the NH and SH (Xie, 2004). Overall, there
are more convective cirrus in the NH ($\sim 60\,\%$) than the SH ($\sim 40\,\%$).

    In contrast to convective cirrus, non-convective cirrus occur most frequently in the boreal winter near the equator (see
Fig. 8b). The seasonal pattern of non-convective cirrus occurrence is negatively correlated with the seasonal pattern of the CPT
temperature. The seasonal cycle of the CPT is driven by the seasonal cycle of stratospheric planetary waves in the extratropical

latitudes (Yulaeva et al., 1994; Highwood and Hoskins, 1998; Jucker and Gerber, 2017). During the boreal winter, stronger
wave activities in the extratropics result in stronger upwelling of the BDC and lower CPT temperatures (Yulaeva et al., 1994;

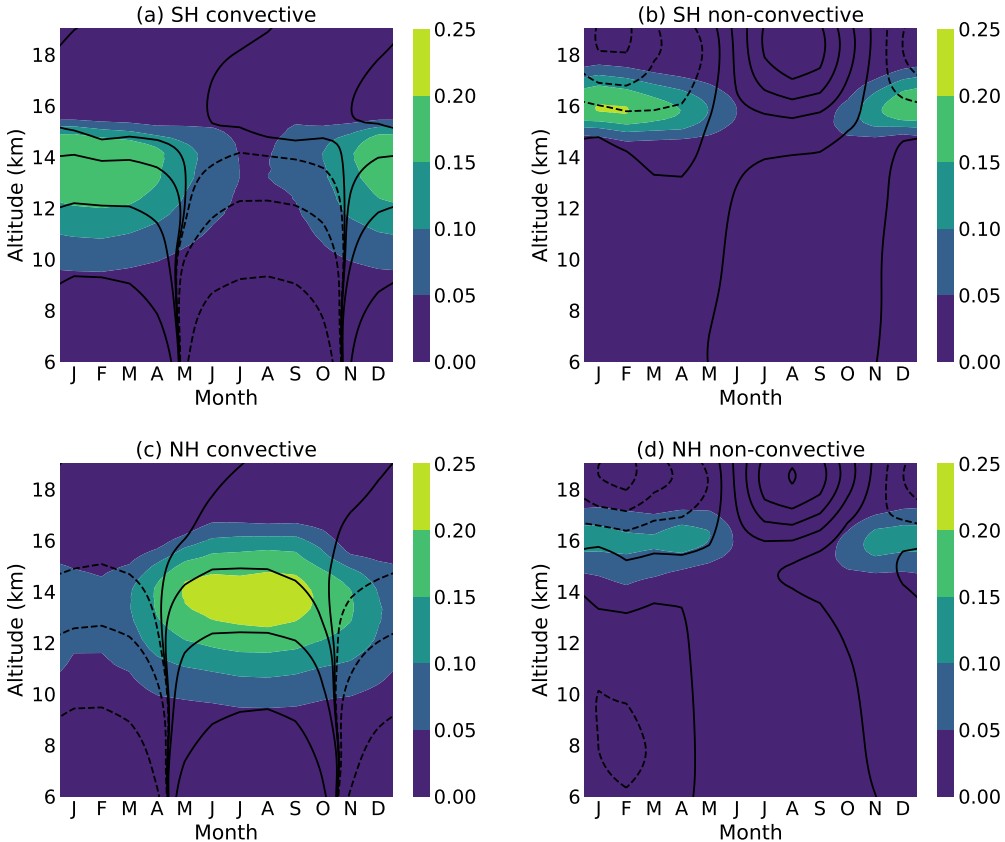

**Figure 7.** Climatological monthly zonal mean frequency of occurrence of convective cirrus (left) and non-convective cirrus (right) over the SH (top) and NH (bottom). Shown with black contours are the monthly SPH perturbations relative to the annual mean SPH (left), and the monthly temperature perturbations relative to the annual mean temperature (right). Positive (negative) SPH and temperature perturbations are shown with solid (dashed) contours.

Holton et al., 1995; Highwood and Hoskins, 1998). In the cold TTL during the boreal winter, local negative temperature perturbations such as those generated by waves can readily increase the RH above the threshold for ice nucleation and so clouds are formed frequently. Figure 8(b) further shows that there are more non-convective cirrus in the SH than the NH. The reason for this is that the maximum center of the upwelling of the BDC is located in the SH in the boreal winter (Mote et al., 1996; Plumb and Eluszkiewicz, 1999).

The seasonal cycles of convective and non-convective cirrus described here are generally consistent with previous studies of cirrus clouds below 14 km–15 km (Sassen et al., 2008; Virts and Wallace, 2010; Nee and Lu, 2021) and cirrus clouds above 14 km–15 km (Tseng and Fu, 2017; Nee and Lu, 2021) respectively. However, the significance here is that we are able to distinguish convective and non-convective cirrus from each other despite the overlapping in their vertical distributions (see Fig. 1b). By separating convective and non-convective cirrus from each other, we can clearly demonstrate the relationships





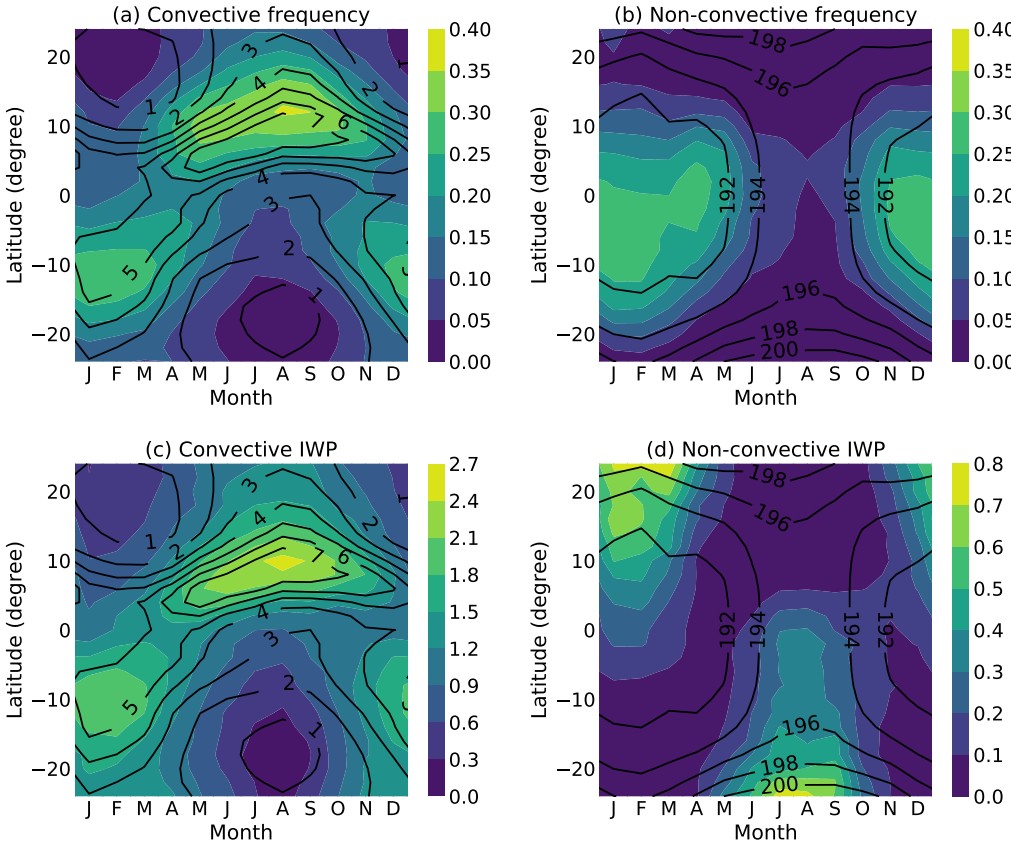

**Figure 8.** Monthly–meridional distributions of the vertical maximum, zonal mean frequency of cirrus occurrence (top), and the zonal mean ice water path (IWP in $g\,m^{-2}$, bottom). The left panels show convective cirrus with the zonal mean precipitation ($mm\,d^{-1}$) in black contours, and the right panels show non-convective cirrus with the zonal mean CPT (K) in black contours.

between convective cirrus and convection, and between non-convective cirrus in the TTL and the temperature there. In previous studies, a particular altitude level (typically around 14 km–15 km) was chosen as a threshold to separate low- and high-altitude cirrus.

### 4.3 Ice water contents

Figure 9 shows the distributions of the occurrence of convective and non-convective cirrus against temperature and in-cloud IWC. The frequency of occurrence of convective cirrus is maximum in the temperature range between 200 K and 250 K, with IWCs between $10^{-3}\,g\,m^{-3}$ and $10^{-1}\,g\,m^{-3}$. In comparison, the histogram of non-convective cirrus shows a distinct maximum count between 190 K and 200 K, which is around the CPT temperature. The IWC of the peak distribution of non-convective
cirrus is between $10^{-5}\,g\,m^{-3}$ and $10^{-3}\,g\,m^{-3}$. However, non-convective cirrus are also occasionally detected below the TTL at



**Figure 9.** Histogram of cloud samples against temperature and in-cloud IWC for (a) convective cirrus and (b) non-convective cirrus.

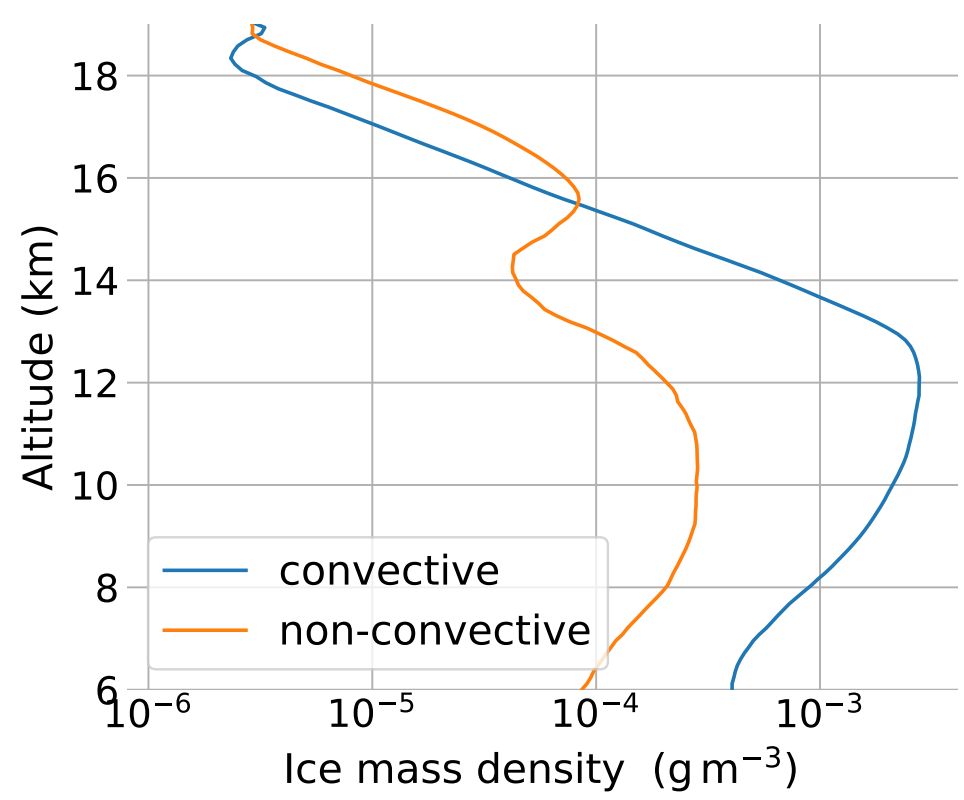

**Figure 10.** Vertical profiles of the ice mass density averaged over the tropical domain due to convective and non-convective cirrus.

temperatures above 200 K (altitudes below 14 km, see also Fig. 1b). These non-convective cirrus at low altitudes have IWCs comparable to those of convective cirrus at the same temperature/altitude levels.

Figure 9 further shows that the IWC in cirrus clouds increases with increasing temperature (decreasing altitude), which is consistent with previous observations (Schiller et al., 2008; Krämer et al., 2016; Heymsfield et al., 2017; Krämer et al., 2020). The behavior of the in-cloud IWC as an increasing function of temperature holds regardless whether the clouds are convective or non-convective cirrus. Thus, non-convective cirrus typically contain less ice water than convective cirrus because non-convective cirrus typically form at lower temperatures (higher altitudes). The different formation mechanisms (convection or non-convective processes) govern the temperature range in which cirrus clouds are formed, through which they govern the IWC of the clouds.

Figure 10 shows that the domain-average ice mass density of non-convective cirrus is about an order of magnitude less than that of convective cirrus throughout most of the atmosphere except above about 15.5 km. The domain-average ice mass density depends on both the in-cloud IWC and the frequency of occurrence of clouds. Given that the IWCs in convective and non-convective cirrus are comparable (at least in the same order of magnitude) at each temperature/altitude level (Fig. 9), the difference in the domain-average ice mass density between convective and non-convective cirrus is determined mainly by the

difference in the frequency of occurrence between the two types of clouds. The 15.5 km level marks the altitude above which the frequency of occurrence (see Fig. 1b) and therefore the domain-average ice mass density of non-convective cirrus exceed those of convective cirrus.

    Finally, Figs. 8(c) and (d) show the seasonal cycle of the ice water path (IWP) in the domain due to convective and non-convective cirrus. The IWP is dominated by the ice water at low altitudes (see Fig. 10). Therefore, the seasonal cycle of the

IWP reflects the seasonal cycle of cirrus clouds at low altitudes. For convective cirrus, the seasonal patterns of the IWP and the maximum frequency of occurrence located at 14 km (see Fig. 1) are similar. This indicates that convective cirrus throughout the troposphere are coupled to each other and to convection. On the other hand, for non-convective cirrus, the seasonal pattern of the IWP is different from that of the occurrence frequency maximum (which is located at 16 km in the TTL, see Fig. 1b). Low-altitude non-convective cirrus are driven by gravity wave activities in the subtropics, which are more prevalent in the

winter months than the summer months in each hemisphere.

## 5   Summary

Based on the anomalies of the SPH and temperature in cloudy air relative to cloud-free air, we have separated the population of tropical cirrus clouds detected by CALIPSO into those of convective origin (convective cirrus) and those of non-convective origins (non-convective cirrus). Convective cirrus occur in moist conditions and include (i) those that form from the freezing

of liquid cloud droplets in convective updrafts, (ii) those that form by in situ ice nucleation from the vapor phase due to the adiabatic or diabatic cooling at the top of deep convection, and (iii) those that form by in situ ice nucleation in the moist air of the convective outflows. Non-convective cirrus occur in dry conditions and form by in situ ice nucleation in response to negative temperature anomalies.

    Convective cirrus tend to occur at lower altitudes than non-convective cirrus. The level of maximum convective cirrus

occurrence is at 14 km, i.e., the bottom of the TTL. In comparison, non-convective cirrus obtain their maximum frequency of occurrence at 16 km. The ratio of the number of convective cirrus to the number of non-convective cirrus is about 3:1 over all altitudes of the tropical atmosphere, 1:1 above 14 km, and 1:3 above the CPT. The majority of non-convective cirrus are located above 14 km, but there are also non-convective cirrus below 14 km. Non-convective cirrus at high altitudes occur near the equator, while non-convective cirrus at low altitudes occur at higher latitudes at the northern and southern edges of the

tropics.

    The seasonal cycle of convective cirrus is consistent with that of tropical convection, while the seasonal cycle of non-convective cirrus above 14 km is consistent with that of the CPT. There are two maxima in the frequency of occurrence of



convective cirrus, one at around 10 °S in the austral summer, and the other at around 10 °N in the boreal summer. In contrast, non-convective cirrus above 14 km occur most frequently near the equator in the boreal winter when the CPT is coldest. Non-
convective cirrus below 14 km occur most frequently in the winter months of each hemisphere whence wave activities are strongest. Overall there are more convective cirrus in the NH than the SH but more non-convective cirrus in the SH than the NH.

The IWC in both convective and non-convective cirrus increases with increasing temperature (decreasing altitude). Thus, non-convective cirrus—which on average occur at lower temperatures (higher altitudes)—tend to have lower IWCs than con-
vective cirrus. However, at a given altitude, the IWCs in convective and non-convective cirrus are comparable to one another (at least in the same order of magnitude). Fresh outflow convective anvil cirrus may have much larger IWCs, but subsequent processes during their life cycles such as cloud horizontal spreading and ice sublimation can decrease the IWCs by several orders of magnitude as shown in previous modeling studies (Dinh et al., 2010, 2012, 2014).

*Author contributions.* Qin Huang carried out the analysis of the data. Both Qin Huang and Tra Dinh contributed to writing the manuscript.

*Competing interests.* No competing interests are present.

*Acknowledgements.* We thank D. Winker for the discussions at the AGU Fall Meeting in 2019, which motivate this work.





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
