# Peer review of "Tropical cirrus clouds of convective and non-convective origins"

_Atmospheric Chemistry and Physics, 2022_

## Referee Comment (RC2)

**Review of "Tropical cirrus clouds of convective and non-convective origins" by D. Huang and Dinh**

The authors use CALIPSO measurements of tropical cirrus, along with ERA5 reanalysis temperature and specific humidity fields, to categorize cirrus as either convective or non-convective. They define convective clouds as those for which the specific humidity is greater than the annual mean at the location where the clouds were observed. As described below, I am skeptical that the paper adds any new understanding of the roles of convection and non-convective processes in governing the occurrence of cirrus clouds, which is the stated goal of the paper. Further, I believe the authors' definition of convective versus non-convective cirrus is misleading because it includes cirrus formed in situ in airmasses with relatively high specific humidity that might (or might not) have been caused by convection somewhere upstream of the observed cirrus.

**Major comments:**

1. As noted by the authors, their definition of "convective-origin" cirrus includes clouds that form well downstream of the convection in airmasses with relatively high specific humidity. This definition is much broader than the conventional definition of convective cirrus, which is limited to cirrus produced directly by the deep convection. Taken to the extreme, the authors' definition could actually include all cirrus in the tropical upper troposphere since deep convection is the primary source of upper tropospheric water vapor. The abstract states that convective cirrus are three times more common than non-convective cirrus. I fear readers will take this statement as the key message of the paper without understanding that the authors' definition of convective cirrus includes any clouds formed downstream of locations where deep convection hydrated the upper troposphere. I believe the authors should choose different terminology to avoid sending a misleading message.

A related issue is that convection is not the only mechanism that can produce specific humidity at a particular location that is higher than the climatological mean. Transport of water from moist to dry regions can also produce a positive humidity anomaly. Further, the mean values used here are apparently annual means (the authors need to be more explicit on this point), in which case the specific humidity anomalies they are using could be dominated by seasonal variability. An alternate approach would be to use regional, seasonal (or subseasonal) mean temperature and humidity based on averages over some nearby domain and limited time period.

2. As noted above, the authors state in the abstract that most tropical cirrus originates from convection. However, their results show a strong height variation in the relative abundance of convective versus non-convective cirrus, with the former dominating below about 15 km, and the latter dominating in the upper TTL. This result is consistent with other data analysis and modeling studies. For example, Jensen et al. (2017) and Schoeberl et al. (2019) used satellite and airborne measurements of clouds and humidity to show that temperature effects dominate in the upper TTL, whereas deep convection controls clouds and humidity in the lower TTL. A number of modeling studies have shown that the observed occurrence frequency and regional distribution of TTL cirrus can be reproduced with only in situ cloud formation (e.g., Ueyama et al., 2015; Ueyama et al., 2018; Schoeberl et al., 2019). Therefore, the conclusion that convection dominates clouds and humidity in the lower TTL, whereas temperature variability dominates cirrus formation in the upper TTL is certainly not new.

As noted by the authors, a number of past studies have shown that TTL cirrus tend to form in anomalously cold airmasses. Therefore, the authors' finding that most cirrus in the TTL occur in times and locations where the temperature is below the climatological mean is no surprise. In general, I do not think the analysis here really clarifies the relationship between convection and

tropical cirrus beyond what was already known.

**Minor comments:**

1. Abstract, lines 8-9: The authors state that "The remaining clouds that are not directly influenced by convection are driven by negative temperature perturbations." As shown later in the paper, this statement is not correct. Some of the clouds with negative SPH perturbations have positive temperature perturbations. In fact, this "unidentified" category comprises 10% of the cloud samples, which is not negligible.

2. Line 32: The authors should cite *Forster and Shine* (2002) since this paper quantified the impact of stratospheric humidity on the radiation budget well before the papers cited.

3. Line 48: When discussing dissipation of convective cirrus, the authors mention cloud horizontal spreading and sublimation, and they cite a series of Dinh et al. papers. However, these papers did not address the evolution of convective cirrus; further, other studies have shown that sedimentation is the dominant process reducing the IWC of convective cirrus *Boehm et al.* (1999); *Jensen et al.* (2018).

4. Lines 60-61: The authors state that "*Ueyama et al.* (2015) calculated backward trajectories that end at the tropopause..." The study actually used curtains along trajectories to simulate TTL cirrus. Further, the authors state that *Luo and Rossow* (2004) "simulated clouds along the trajectories." As far as I can tell, this paper did not use cloud simulations. They simply tracked the clouds in the satellite imagery.

5. Lines 68-69: In addition to the papers cited here, *Jensen and et al.* (2020) recently documented cases of convective hydration of the lower stratosphere.

6. Lines 117-121: The mean specific humidity and temperature are apparently averaged over the entire data time period (although this is not stated explicitly). This would mean the averages are climatological, annual means, and the resulting correlations between cloud occurrence, humidity anomalies, and temperature anomalies will largely just represent the seasonal variations in these quantities. I think it would make more sense to use seasonal means for the clear-sky averages.

7. Lines 171-177: The authors note that the peak of the non-convective cloud frequency occurs below the CPT, and as an explanation, they cite the modeling study indicating that radiatively-driven circulations will be damped if the temperature in the cloud increases with height. First, there is no observational evidence showing that radiatively-driven circulations routinely occur in cirrus near the tropopause, and the lifetime of TTL cirrus (limited by wave-driven temperature perturbations) is typically too short for these circulations to develop (*Jensen et al.*, 2011). Second, there is a much simpler explanation for the peak cloud frequency altitude occurring below the tropical mean CPT: The CPT altitude varies considerably from location to location and time to time. Therefore, at the mean CPT altitude, you are often above the local CPT, in which case cloud formation would be suppressed. As shown by *Pan and Munchak* (2011), TTL cirrus cloud tops generally occur just below the local CPT.

8. Lines 185-192: The authors use their convective occurrence height distribution to estimate the frequency of overshooting above the CPT. I do not believe there is any quantitative value to this calculation. First, it is entirely possible that the few convective clouds (with positive SPH anomalies) observed above the CPT are just places where the SPH is above the climatological mean with no relationship to deep convection. Second, as noted above, the CPT height varies spatially and temporally. Therefore, for this calculation to have any value, the authors would need

to carefully determine which of the clouds are above the *local* CPT.

**References**

Boehm, M. T., J. Verlinde, and T. P. Ackerman (1999), On the maintenance of high tropical cirrus, *J. Geophys. Res.*, *104*, 24,423–24,433.

Forster, P. M. F., and K. P. Shine (2002), Assessing the climate impact of trends in stratospheric water vapor, *Geophys. Res. Lett.*, *29*, 10–1/4.

Jensen, E. J., and et al. (2020), Assessment of observational evidence for direct convective hydration of the lower stratosphere, *J. Geophys. Res.*, *125*, doi:https://doi.org/10.1029/2020JD032793.

Jensen, E. J., L. Pfister, and O. B. Toon (2011), Impact of radiative heating, wind shear, temperature variability, and microphysical processes on the structure and evolution of thin cirrus in the tropical tropopause layer, *J. Geophys. Res.*, *116*, doi:1029/2010JD015,417.

Jensen, E. J., S. C. van den Heever, and L. D. Grant (2018), The lifecycles of ice crystals detrained from the tops of deep convection, *J. Geophys. Res.*, *123*, in press.

Luo, Z., and W. B. Rossow (2004), Characterizing tropical cirrus life cycle, evolution, and interaction with upper-tropospheric water vapor using lagrangian trajectory analysis of satellite observations, *J. Atmos. Sci.*, *17*, 4541–4563.

Pan, L. L., and L. A. Munchak (2011), Relationship of cloud top to the tropopause and jet structure from calipso data, *J. Geophys. Res.*, *116*(D12201), doi:10.1029/2010JD015,462.

Ueyama, R., E. J. Jensen, L. Pfister, and J.-E. Kim (2015), Dynamical, convective, and microphysical control on wintertime distributions of water vapor and clouds in the tropical tropopause layer, *J. Geophys. Res.*, *120*, doi:10.1002/2015JD023,318.

---

## Author Comment (AC1)

**Response to Reviewer 1**

**1 Overall assessment**

***Reviewer*** — The manuscript uses 9 years of CALIPSO level 3 gridded monthly data to separate the tropical cirrus clouds into those from convective and non-convective origin. The authors define clouds associated with positive specific humidity anomalies as convective origin and clouds associated with negative temperature anomalies as non-convective origin cirrus. While their method seems overly simplistic at first sight, their robust, physically justifiable results speak for themselves and helped overcome my initial skepticism about the use of very coarse time and spatial resolution of the satellite dataset. This is a nice and clear study, and I recommend publication after the listed comments are addressed.

***Authors*** — We thank the reviewer for his encouraging feedback and for the helpful questions and comments below. Prompted by the feedback from both reviewers, we have made changes to the manuscript and believe it has been significantly improved as a result. The list of major changes to the manuscript is attached. It may be helpful for the readers to refer to the list of changes before reading the discussions below. We are also attaching two versions of the revised manuscript, one with tracked changes and one without tracked changes. The discussions below refer to the line numbers in the version with tracked changes. Please find below our point-by-point reply, first to the general comments, followed by the minor comments of the reviewer.

**2 General comments**

1. ***Reviewer*** — With CALIPSO, you are limited to clouds with optical thickness smaller than about 3. Is this a significant limitation of the study? What proportion of the clouds is missed?

   ***Authors*** — We are indeed limited to clouds with optical depths (ODs) smaller than about 3 with CALIPSO. This is not a significant limitation of the study because only a small or negligible number of clouds are missed as there are few cirrus clouds with ODs greater than 3. CALIPSO user guide shows that cirrus cloud frequency decreases exponentially with increasing OD for OD larger than 0.2 and that more than 90 % of cirrus clouds have ODs less than 1 (see `https://www-calipso.larc.nasa.gov/resources/calipso_users_guide/data_summaries/profile_data.php`). Other independent measurements also confirm that the majority of cirrus clouds are optically thin with ODs less than 3. For example, ground-based lidar measurements showed that, over the tropical Amazonia region, cirrus clouds with ODs less than

0.3 account for about 80 % of all cirrus clouds; the remaining 20 % are cirrus with ODs greater than 0.3 (Gouveia et al., 2017). In another study by Kumar and Venkatramanan (2020) using a ground-based Mie lidar, cirrus clouds with ODs greater than 0.3 were found to account for only 14 % of cirrus clouds observed over Gadanki, India. From these numbers and the exponential decrease of cirrus cloud frequency with increasing OD, we estimate that cirrus clouds with ODs greater than 0.3 account for about 10–20 % of tropical cirrus clouds, and cirrus clouds with ODs greater than 3 probably account for less than a few percent of tropical cirrus clouds.

2. *Reviewer* — Is it fair to say that a positive specific humidity anomaly must be associated with convection? What if convection with relative humidity with respect to ice of 100 % reaches an ice supersaturated region? There is ample evidence that deep convection on average hydrates the upper troposphere, but I think the authors should nevertheless discuss the other possibility and how it could influence their results.

   *Authors* — It is true that some convection events can produce negative specific humidity (SPH) anomaly. For this reason and the reasons pointed out by the second reviewer, we have changed the terminology used in the manuscript, from cirrus that originate from convection and non-convective processes to moist and dry cirrus, respectively. The new names refer to how we classify the clouds based on the SPH and temperature anomalies. We no longer assume a priori that convection always leads to hydration. The new terminology does not change the main results of the analysis, i.e., that the monthly spatiotemporal distribution of moist cirrus (formerly convective cirrus) is consistent with that of convection, while the monthly spatiotemporal distribution of dry cirrus (formerly non-convective cirrus) is distinct from that of convection. The latter result indicates that convection events that lead to negative SPH anomaly do not happen frequently enough to show up in the monthly data. The monthly seasonal patterns of both high-altitude dry cirrus (Fig. 8b) and low-altitude dry cirrus (Fig. 8d) are distinct from that of convection. In other words, on the monthly and seasonal time scales, the population of dry cirrus is not driven by convection.

3. *Reviewer* — Could you verify your cloud classification method on a subset of instantaneous CALIPSO profile data? Would the results based on instantaneous data agree with the gridded, 1-monthly data?

   *Authors* — We have verified the method with the CALIPSO Level 2 cloud pro-file dataset v4.20 (`https://asdc.larc.nasa.gov/project/CALIPSO/CAL_LID_L2_05kmCPro-Standard-V4-20_V4-20`) for January 2015. The time resolution of the CALIPSO Level 2 data is 0.74 s. Due to limited computer resources, we cannot process the data at this resolution. Therefore, we averaged the Level 2 data over time to obtain the daily data for January 2015. The daily meteorological conditions were obtained from ERA5 for the same month and used for the classification of clouds in the daily-averaged Level 2 data. Figure 1 shows that the occurrence of moist and dry cirrus is qualitatively consistent between CALIPSO Level 2 and Level 3 datasets.

**3 Specific comments**

1. *Reviewer* — Abstract: For clarity, I suggest avoiding the use of abbreviations in the abstract (unless strictly needed).

[Figure]

Figure 1: Latitude–altitude profiles of the frequency of occurrence of moist cirrus (left) and dry cirrus (right) in January 2015, derived from CALIPSO Level 3 Ice Cloud Product (top) and CALIPSO Level 2 Ice Cloud Product (bottom).

*Authors* — We agree and have removed all abbreviations from the abstract.

2. *Reviewer* — Introduction: I'm missing a few more lines describing why it is important to separate the origin of cirrus. In principle, the models could simulate the correct cloud amount and cirrus properties even without correctly accounting for their origin.

   *Authors* — We have revised the introduction to discuss that our motivation is the occurrence of cirrus clouds in response to the SPH and temperature anomalies. The origins of cirrus clouds are inferred from the corresponding analysis for academic interests, while the analysis of cloud occurrence in response to the SPH and temperature anomalies has practical modeling applications. The sentence "The purpose of this paper is to quantify the roles of convection and non-convective processes in governing the occurrence of cirrus clouds." has been removed (please see lines 33–34). To clarify the applications in modeling practice, we added the following text to the end of the manuscript (lines 407–414):

   "The method proposed here to study cirrus clouds can be applied in model development to improve the representation of cirrus clouds in numerical simulations. We have demonstrated that the spatiotemporal distribution of cirrus clouds is governed by the SPH, temperature, and their variations. Therefore, models would need to accurately represent the SPH, temperature, and their variances in order to accurately simulate the distribution of cirrus clouds. It would be useful to compare between observations and numerical simulations in terms of the frequency and magnitude of the moisture and temperature anomalies and how they affect the occurrence of cirrus clouds. Such a comparison would reveal the specific strategies on how to adjust the model parameterization schemes (e.g., the convection scheme, the gravity wave drag scheme, and/or the microphysics scheme) to improve the representation of cirrus clouds in models."

3. *Reviewer* — Line 26: Li et al., 2012 (doi: 10.1029/2012JD017640) may be a good reference about the uncertainties in cirrus, at least with respect to the ice water content.

   *Authors* — Li et al. (2012) is now cited on line 32.

4. *Reviewer* — Lines 46–47: The sentence starting with "Wang and Dessler" is missing something.

   *Authors* — We added "For example, " before Wang and Dessler (2012) on line 52.

5. *Reviewer* — Line 48: It may be appropriate to add references explicitly looking at the decay of convective origin clouds. If I am not mistaken, the cited papers all refer to the evolution of in-situ TTL cirrus.

   *Authors* — Thank you for letting us know about these relevant papers. The papers by Gehlot and Quaas (2012) and Gasparini et al. (2021) on the decay of convective-origin cirrus clouds are now cited on lines 56–57 and 405–406.

6. *Reviewer* — Lines 59–62: I would suggest also mentioning studies using high cloud trajectories in climate models, e.g., Gehlot and Quaas, 2012 (doi: 0.1175/JCLI-D-11-00345.1) and Gasparini et al., 2021 (doi: 10.1175/JCLI-D-11-00345.1).

   *Authors* — Here we discuss how Lagrangian trajectory calculations were used to distinguish cirrus of convective and non-convective origins. On the other hand, Gehlot and Quaas (2012) and Gasparini et al. (2021) focused on convective-origin cirrus clouds only. Therefore, we reference

these two papers elsewhere (lines 56–57 and 405–406). The paper by Luo and Rossow (2004) which was cited here originally has also been moved to lines 56–57 and 405–406 where processes that affect the decay of cirrus clouds are discussed.

7. ***Reviewer*** — Lines 90–92: Does your method work also for regions with a limited annual cycle of convection, e.g., for parts of the tropical western Pacific?

   *Authors* — Yes, our method works there.

8. ***Reviewer*** — Section 3: How is cloud fraction defined? Can it be only 0 or 1 or is it also expressed as a fraction? If fractions are used, how do you consider them in the analysis of in-cloud vs clear-sky grid boxes?

   *Authors* — The ice cloud fraction (ICF) is calculated by,

   $$\mathrm{ICF} = \frac{ice\_cloud\_accepted\_samples}{cloud\_accepted\_samples + cloud\_rejected\_samples + cloud\_free\_samples},$$

   (see `https://www-calipso.larc.nasa.gov/resources/calipso_users_guide/qs/cal_lid_l3_cloud_occurrence_v1-00.php`). The ICF varies between 0 and 1. At a given time, a grid box is considered cloudy if the ICF is greater than 0.01. We have added the definition of cloudy grid boxes to lines 140–142.

9. ***Reviewer*** — Figure 3: I would suggest using a symlog scaling (`https://matplotlib.org/stable/gallery/scales/symlog_demo.html`), so that one can see more than just the temperature-dependent increase in Dq (i.e., basically Clausius–Clapeyron). If you use matplotlib for plotting, this is how you could do it: plt.gca().set–yscale('symlog', linthreshy=1e-2).

   *Authors* — Thanks for this suggestion. The figure is now plotted in logarithmic scale.

10. ***Reviewer*** — Figure 5: density = IWC, right? Please, be consistent. Caveat: CALIPSO lidar will not penetrate into optically thick clouds, so the lower part of the plot is biased to low IWC.

    *Authors* — We have changed the word 'density' into 'IWC' for consistency in the revised manuscript. Although CALIPSO retrieval of IWC may not be perfect, it was found to be in good agreement with the radar-lidar retrieval over Darwin, Australia (Protat et al., 2010).

11. ***Reviewer*** — Lines 209–218: Please, don't use parentheses to indicate the opposite of an idea. This makes the text really hard to understand. See also `https://eos.org/opinions/parentheses-are-are-not-for-references-and-clarification-saving-space`.

    *Authors* — We have rewritten these so that parentheses are no longer used. Please see lines 281–289.

12. ***Reviewer*** — Figure 10: Could you explicitly mention already in the caption that you show an "all-sky" average?

    *Authors* — Figure 10 now shows the profiles of both the grid-average IWC and in-cloud IWC. The caption refers to the "grid-average" (instead of "all-sky") IWC so as to be consistent with CALIPSO terminology (see `https://www-calipso.larc.nasa.gov/resources/calipso_users_guide/qs/cal_lid_l3_ice_cloud_v1-00.php`).

13. *Reviewer* — Line 263: It's really hard to see from Fig. 9 if the average/median IWC of convective and non-convective cirrus are comparable in a given temperature bin or not. Could you for example express it in numbers/median values? Also, the IWC is certainly biased low at $T > 240\,\mathrm{K}$ due to the limitations of the CALIPSO lidar measurements.

*Authors* — We added the vertical profiles of the in-cloud IWC in moist and dry cirrus to Fig. 10. These profiles show more clearly (than Fig. 9) that the average IWCs in moist and dry cirrus are comparable. The biases in IWCs would affect moist and dry cirrus equally and so they would not alter our conclusion that the IWCs moist and dry cirrus are comparable for a given temperature/altitude bin.

14. *Reviewer* — Line 303: As in the introduction, it may be worthwhile to add some citations that actually studied the decay of convective-origin cirrus, and not only the in-situ generated cirrus.

*Authors* — We now cite Gehlot and Quaas (2012) and Gasparini et al. (2021) here. Please see line 405–406.


90  suggest that moist cirrus are driven by convection. In contrast, the spatiotemporal distribution of dry cirrus is distinct from that of tropical convection, suggesting that these clouds are unrelated to convection. The seasonal cycle of

 dry cirrus in and above the TTL is consistent with that of  the cold point tropopause (CPT) temperature, and the seasonal cycle of  dry cirrus below the TTL is consistent with that of  wave activities in the troposphere.

The knowledge that tropical cirrus clouds are driven by convection as well as by negative temperature anomalies associated with non-convective processes is not new (see the references cited above). However, to the best of our knowledge, this is the first time that individual vertical profiles of moist cirrus (that are related to convection) and dry cirrus (that are unrelated to convection) are obtained from observational data. In previous observational studies, a particular altitude (typically around 14 km–15 km, the bottom of the TTL) was chosen as a threshold level to separate cirrus driven by convection at lower altitudes and those which are driven by negative temperature anomalies at higher altitudes. Such a clear-cut separation may not be appropriate as the vertical profiles of moist and dry cirrus that we obtain here suggest that convection and non-convective sources contribute almost equally to the total occurrence of cirrus clouds above 14 km. Furthermore, the vertical profile of dry cirrus reveals that a significant number of cirrus clouds in the troposphere are formed by negative temperature anomalies unrelated to convection. Much of previous research on the effect of temperature anomalies on cirrus clouds has focused on the higher altitudes in and above the TTL only. The other significant finding is that moist and dry cirrus contain similar IWCs if they are located at the same altitude or temperature level. Previously, cirrus that originate from convection were often thought to have larger IWCs [see e.g. *Wang and Dessler*, 2012].

The remaining of the paper is organized as follows. The data and methodology are described respectively in Sections 2 and 3. Section 4 discusses the characteristics of the occurrence of  moist and dry cirrus in the tropics, including their spatiotemporal distributions and IWCs. Section 5 contains the summary.

**2 Data**

We analyze the monthly-mean, three-dimensional cirrus cloud occurrence and IWC of the Lidar Level 3 Ice Cloud Data, Standard Version 1-00 [*NASA Langley Atmospheric Science Data Center*, 2018]. This data was collected with the Cloud-Aerosol LIdar with Orthogonal Polarization (CALIOP) instrument on  CALIPSO [*Winker et al.*, 2010]. CALIOP is capable of detecting clouds with optical depths of 0.01 or less [*Winker et al.*, 2007]. For this reason, CALIOP data are well suited for studies of cirrus clouds, many of which are optically thin. The user guides for the monthly Lidar Level 3 Ice Cloud Data can be found online (see https://www-calipso.larc. nasa.gov/resources/calipso_users_guide/qs/cal_lid_l3_ice_cloud_v1-00.php and https://www-calipso.larc.nasa.gov/resources/ calipso_users_guide/qs/cal_lid_l3_cloud_occurrence_v1-00.php).

The spatial resolution of the monthly Lidar Level 3 Ice Cloud Data is 2.5° in the zonal direction, 2.0° in the meridional direction, and 120 m in the vertical. The monthly data is not suitable to conduct case study of clouds that occur in response to individual convection or wave events. However, if convection and wave activities have intrinsic seasonal cycles, their effects on the seasonal cycles of cirrus occurrence should be captured in the monthly data. Thus, this data is appropriate for us to study the

[Figure]

**Figure 1.** Climatological mean frequency of occurrence of cirrus clouds in the tropics: (a) latitude–altitude profile of the zonal mean frequency of occurrence, and (b) vertical profile of the frequency of occurrence averaged over the tropics.

spatial distribution of cirrus clouds on the seasonal and climatological time scales. Figure 1 shows the frequency of occurrence of cirrus clouds in the tropics between 24 °S and 24 °N over the 9-year period from January 2007 to December 2015 obtained from CALIPSO. The overall spatial distribution of cirrus clouds in the figure is consistent with previous observational studies using CALIPSO [*Sassen et al.*, 2008; *Mace et al.*, 2009; *Hong and Liu*, 2015] and satellite radiometers [*Wang et al.*, 1996].

 For the meteorological conditions surrounding cirrus clouds, specifically the specific humidity (SPH) and temperature of the atmosphere, we use the ERA5 data [*Hersbach et* al., 2020]. In addition, we obtain the data for precipitation from the Version-2 Global Precipitation Climatology Project (GPCP) Monthly Precipitation Analysis [*Adler et al.*, 2003]. The  SPH, temperature, and precipitation data were downloaded for the same temporal and spatial domains as the ice cloud data above, and then they were interpolated to the same grid as the ice cloud data.

**3 Identifying  moist and dry cirrus**

Figure 2 shows the vertical profiles of the relative humidity (RH) that has been averaged over time and the tropical domain in cloudy and  all-sky conditions. The RH discussed in this work is specifically with respect to ice. At a given grid box location,  cloudy conditions refer to the times when cirrus clouds are detected, i.e., when the ice cloud fraction is greater than 0.01, in the grid box. The figure shows that on average the RH is greater in cloudy conditions than in  all-sky conditions at every altitude. This is consistent with existing observations [*Sandor et al.*, 2000; *Kahn et al.*, 2008, 2009; *Krämer et al.*, 2020] that the RH is greater in cloudy conditions to support the formation and maintenance of  clouds.

[Figure]

**Figure 2.** Vertical profiles of the climatological  mean RH in cloudy and  all-sky conditions .

The RH is related to the temperature ($T$) and the SPH ($q$) via

$$\text{RH} = \frac{p_v}{e_{si}(T)} = \frac{R_v}{R}\frac{q}{e_{si}(T)}p, \tag{1}$$

where $R = 287\,\text{J}\,\text{kg}^{-1}\,\text{K}^{-1}$ and $R_v = 461\,\text{J}\,\text{kg}^{-1}\,\text{K}^{-1}$ are respectively the specific gas constants of air and water vapor, $p$ is atmospheric pressure, $p_v$ is the partial pressure of water vapor in air, and $e_{si}(T)$ is the saturation water vapor pressure with respect to ice, a function of temperature. The function $e_{si}(T)$ increases with temperature and is calculated based on the empirical formula given by *Murphy and Koop* [2005]. According to Eq. (1), at a given pressure level, large RH values inside clouds relative to  the all-sky condition must arise from positive SPH anomalies and/or negative temperature anomalies.

 Let $\Delta q(x,y,z,t) = q_{cld}(x,y,z,t) - \overline{q}(x,y,z)$ denote the difference between the SPH in the cloud sample at the location $(x,y,z)$ and time $t$ and the  time-average SPH at the same location. The  time-average SPH ($\overline{q}$) is obtained by averaging the SPH over  all the times in the dataset regardless of whether clouds are present in the grid box. Similarly,  $\Delta T(x,y,z,t) = T_{cld}(x,y,z,t) - \overline{T}(x,y,z)$ is the difference between the temperature in the cloud sample at the location $(x,y,z)$ and time $t$ and the  time-average temperature at the same location. The vertical profiles of the climatological mean, tropical average $\Delta q$ and $\Delta T$ are shown in Fig. 3. The figure shows that $\Delta q$ (green line) is positive in the troposphere, indicating that most cirrus clouds in the troposphere are formed and maintained in the months of positive SPH anomalies. Furthermore, $\Delta q$ decreases exponentially with altitude, consistent with the fact that the background SPH decreases exponentially with altitude. On the other hand, the magnitude of the temperature anomalies experienced by cirrus clouds is small in most of the troposphere, and it becomes significant only above 14 km or so. The result that cirrus clouds above 14 km experience significant negative temperature anomalies is consistent with previous studies [*Boehm and Verlinde*, 2000; *Virts et al.*, 2010; *Virts and Wallace*, 2010; *Tseng and Fu*, 2017].

[Figure]

**Figure 3.** Vertical profiles of the climatological mean, tropical average differences in (a) SPH and (b) temperature between cloudy and  all-sky conditions.

 Let us refer to the clouds in which $\Delta q > 0$ as moist cirrus. We expect that moist cirrus are influenced by convection since positive SPH anomalies are largely produced by the upward transport of moisture by convection in the free atmosphere in the tropics. Some positive SPH anomalies may be located away from active convection, but even in these cases the source of moisture must be the convective outflows that have been transported horizontally by the winds [see *Salathé and Hartmann*, 1997; *Sohn et al.*, 2008; *Das et al.*, 2011].  Moist cirrus include clouds that form within the convective updrafts and at the top of convection, as well as those that form in the moist air of the convective outflows downstream of convection. Examples of the latter type of  moist cirrus were recently reported by *Cairo et al.* [2021]. Using this  definition, we find that  58 % of tropical cirrus clouds are  moist cirrus (see Fig. 4). About  60 % of moist cirrus experience positive temperature anomalies, and the  remaining 40 % experience negative temperature anomalies.

The remaining cirrus clouds in which $\Delta q \leq 0$ consist of two categories. The first category comprises of the cloud samples in which $\Delta q \leq 0$ and $\Delta T < 0$.  We refer to these clouds as dry cirrus. Dry cirrus make up 34 % of tropical cirrus clouds (see Fig. 4).  Since convection sometimes leads to dehydration of the air [*Jensen et* al., 2007], in principle some dry cirrus can be associated with convection. However, our analysis (see Section 4.2) shows that the monthly spatiotemporal distribution of dry cirrus is distinct from those of convection and moist cirrus. The maximum frequency of occurrence of dry cirrus is located remotely away from the maximum frequency of occurrence of moist cirrus. Thus, the negative temperature anomalies that form and maintain  dry cirrus are unlikely to be the cooling at the top of convection. Rather, they are associated with waves and/or the adiabatic cooling associated with the upwelling of the BDC.

[Figure]

**Figure 4.** Histogram of cloud samples against the differences in SPH and temperature between cloudy and all-sky conditions.

 The last category of clouds is for those in which $\Delta q \leq 0$ and $\Delta T \geq 0$. These clouds are  driven by neither positive SPH anomalies nor negative temperature anomalies. They are most likely in the decaying stage of their lifetimes. In these cases, the SPH and temperature anomalies cannot be used to identify their formation and maintenance mechanisms.

 Decaying clouds make up 8 % of tropical cirrus ~~clouds, the method described above allows us to identify the majority of cirrus clouds and their relationship with convection. Furthermore, as shown in Section 4 below, the spatial distributions and seasonal cycles of convective and non-convective cirrus are distinct from each other. The occurrence of convective cirrus is consistent with the location and the seasonal cycle of tropical convection, while the occurrence of non-convective cirrus is consistent with the CPT. The seasonal cycle of the CPT is strongly coupled to that of the BDC *Highwood and Hoskins*, 1998; *Jucker and Gerber*, 2017. These results suggest that the method we propose is appropriate to separate convective and non-convective cirrus~~ (see Fig. 4).

**4 Characteristics of the occurrence of  moist and  dry cirrus**

**4.1 Spatial distributions**

Figures 1 and 5  show that the frequency of occurrence of  moist cirrus is maximum  near 14 km ($\sim$ 150 hPa), coincided with the level of zero net radiative heating rate, which is often defined as the bottom of the TTL [*Fueglistaler et al.*, 2009]. The 14 km altitude is also approximately the level of neutral buoyancy, which provides the upper bound for convective development in the vertical [*Takahashi and Luo*, 2012]. The level of maximum convective mass outflow is located several kilometers lower at around 10 km–11 km [*Takahashi and Luo*, 2012].  Moist cirrus between the level of neutral buoyancy (14 km) and the level of maximum convective outflow (10 km–11 km) are likely anvil cirrus.  Moist cirrus above 14 km are likely to originate from (i) the further lofting, spreading and detachment of anvils, (ii) in situ ice

[Figure]

**Figure 5.** (a) Latitude–altitude profiles of the climatological zonal mean frequency of occurrence filled colored contours) and  dry cirrus (white contours), and (b) vertical profiles of  the climatological zonal mean frequency of occurrence of moist and dry cirrus averaged over the SH (dashed) and  NH (solid).

nucleation in the moist air of the convective outflows in response to cold anomalies (see Fig. 3) associated with the cooling at the top of deep convection and/or waves. At lower altitudes (below 10 km or at temperatures above 235 K),  moist cirrus originate from mixed-phase clouds [*Heymsfield et al.*, 2017], i.e. they are of liquid origin [terminology following *Krämer et al.*, 2016].  More moist cirrus experience positive temperature anomalies than negative temperature anomalies (see Figs. 3 b and 4). The positive temperature anomalies likely arise from the latent heat release in convection.

 Dry cirrus tend to occur at higher altitudes than  moist cirrus. The frequency of occurrence of  dry cirrus maximizes at around 16 km, below the CPT (see Figs. 1 and 5). The climatological tropical mean CPT is found to be at 16.8 km. The level of maximum cirrus occurrence is capped above by the CPT potentially because of two reasons. Firstly, the RH decreases with altitude above the CPT as temperature increases with altitude (see Fig. 2). Thus, above the CPT the negative temperature perturbations must be of large magnitudes to raise the RH above the threshold of ice nucleation. Secondly, the modelling study by *Dinh et al.* [2010] suggested that a necessary condition for cirrus clouds to self-maintain for a long time is that the temperature in the cloud layer decreases with altitude. In this situation, the circulation induced by the cloud radiative heating produces in-cloud water vapor flux convergence that acts against ice sublimation [*Dinh et al.*, 2010]. On the other hand, when the temperature in the cloud layer increases with altitude (such as above the CPT), the circulation induced by the cloud radiative heating produces in-cloud water vapor flux divergence that enhances ice sublimation.  As a result, clouds above the CPT are short-lived and  so the frequency of cloud occurrence above the CPT is small.

**Table 1.** Percentage contributions of the different types of cirrus clouds to the total cirrus occurrence in different layers of the tropical atmosphere. The bottom of the tropical tropopause layer (TTL) is located near 14.0 km ($\sim$ 150 hPa). The climatological tropical mean CPT is at 16.8 km ($\sim$ 100 hPa).

| |  Moist |  Dry |  Decaying |
|---|---|---|---|
| Above CPT |  22 | 76 |  2 |
| Above 16.8 km | 25 | 73 | 2 |
| Above 14.0 km |  48 |  46 | 6 |
| Below 14.0 km | 67 | 26 | 7 |
| All altitudes |  58 |  34 |  8 |

Table 1 shows the percentage contributions of the different types of cirrus clouds to the total cirrus in different layers of the tropical atmosphere. The table shows that  moist cirrus dominate the entire atmosphere and the troposphere below 14 km, i.e., the bottom of the TTL. Above 14 km,  moist and dry cirrus contribute almost equally to the total cirrus cloud occurrence . This is consistent with the observational study by *Massie et al.* [2002] that convection affects half of cirrus population in the TTL, although they studied cirrus clouds over the maritime continent only. This result is also consistent with the modeling study by *Schoeberl et al.* [2018] in which cirrus cloud fraction in the TTL doubles when convection is included in the model simulations. Above the CPT,  dry cirrus make up the large majority (76 %) of clouds, but the percentage of  moist cirrus is not negligible ( 22 %). Similar numbers (73 % dry cirrus and 25 % moist cirrus) are obtained above 16.8 km, which is the altitude of the climatological tropical mean CPT .

~~Based on the vertical profile of the frequency of occurrence of convective cirrus clouds (see Fig. 1b) , we can estimate the degree of overshooting convection above the CPT. Above the CPT, the frequency of occurrence of convective cirrus decreases with altitude, indicating that the degree of penetration of convection into the stratosphere decreases with altitude. At the CPT, the frequency of occurrence of convective cirrus is 1.7 %. This provides the upper bound for the occurrence of overshooting convection injecting ice into the stratosphere because not all convective cirrus are formed within the convective updrafts; some convective cirrus are formed in situ in the moist air of the convective ouflows. Gettelman et al. [2002] found based on cloud brightness temperatures that convection is present above the CPT about 0.5 % of~~

Figure 5 shows that there are more moist cirrus clouds in the Northern Hemisphere (NH) than in the Southern Hemisphere (SH) at every altitude. This is consistent with the fact that convection is tronger in the NH than in the SH. On the other hand, above 15 km there are less dry cirrus clouds in the NH than in the SH, and below 15 km there are more dry cirrus clouds in the NH than in the SH. Above 15 km, the occurrence of dry cirrus is anti-correlated with the CPT temperature, which is

[Figure]

**Figure 6.** Latitude–altitude profiles of the zonal mean frequency of occurrence of moist cirrus (filled colored contours) and dry cirrus (white contours) in December–January–February (DJF), March–April–May (MAM), June–July–August (JJA), and September–October–November (SON).

The meridional pattern of convective cirrus occurrence is bimodal and asymmetric about the equator (Figs. 5a and **??**). There are two maxima at approximately 10 °S and 10 °N, with the northern hemisphere (NH ) maximum being larger than the southern hemisphere (SH) maximum, consistent with the fact that convection is stronger in the NH. In comparison, the meridional pattern of non-convective cirrus occurrence is unimodal, with the maximum frequency of occurrence centered around the equator (Figs. 5b and **??**). The different spatial distributions of convective and non-convective cirrus suggest that the mechanisms governing the occurrence of non-convective cirrus is distinct from convection . This topic is further discussed in Section 4.2.

Figures 5(c) and  (d) show the grid-average ice mass density associated with convective and non-convective cirrus. For both types of clouds controlled by the upwelling of the BDC. The maximum center of the upwelling of the BDC is located in the SH [*Mote et* al., 1996; *Plumb and Eluszkiewicz, 1999*], so there are more dry cirrus in the SH above 15 km. To explain why below 15 km there are more dry cirrus clouds in the NH than the SH, the maximum ice mass density is located below we refer to Fig. 6

which shows the occurrence of moist and dry cirrus over the four seasons. In  NH, dry cirrus below 15 km occur most frequently in the boreal winter (December–January–February) and to a lesser extent in the boreal spring (March–April–May). In these months of the year, convection is the least active in the NH and the most active in the SH, consistently with the distribution of moist cirrus clouds shown in Fig. 6. In other words, dry cirrus below 15 km are located remotely away (in the opposite hemisphere) from the most active convection. It is thus unlikely that dry cirrus clouds below 15 km are driven by convection, even though convection can produce negative temperature anomalies. We posit that these low-altitude dry cirrus are instead driven by the temperature anomalies associated with gravity waves in the troposphere. Observations of gravity waves in the tropical troposphere are rare. However, the particular radiosonde data over the tropical regions of the USA [*Zhang et al.*, 2010] indeed show that the energy density of tropospheric gravity waves is maximum in the boreal winter–spring, consistent with the occurrence of low-altitude dry cirrus in the NH in these seasons.

**4.2 Seasonal cycles**

Figure 7 shows how the seasonal cycle of  moist cirrus is forced from the bottom up by the seasonal cycle of the SPH in the troposphere, while the seasonal cycle of  dry cirrus in the TTL is forced from the top down by the seasonal cycle of the temperature in the TTL and lower stratosphere. In each hemisphere,  moist cirrus occur most  frequently during the summer  months when the SPH perturbations are positive and least frequently during the winter months when the SPH perturbations are negative. The seasonal cycle of  moist cirrus in the NH is thus opposite of that in the SH, i.e., when the frequency of occurrence of  moist cirrus is maximum in the NH, it is minimum in the SH.  In contrast, in both the NH and SH  dry cirrus occur most frequently in the boreal winter when the temperatures in the TTL are minimum  and least frequently in the boreal summer when the temperatures in the TTL are maximum.

~~Climatological monthly zonal mean frequency of occurrence of convective cirrus (left) and non-convective cirrus (right) over the SH (top) and NH (bottom). Shown with black contours are the monthly SPH perturbations relative to the annual mean SPH (left), and the monthly temperature perturbations relative to the annual mean temperature (right). Positive (negative) SPH and temperature perturbations are shown with solid (dashed) contours.~~

Figure 8(a) shows the seasonal migrations of  moist cirrus and precipitation between the NH in the boreal winter and the SH in the austral summer. The similar seasonal patterns of  moist cirrus and precipitation suggest that these

[Figure]

**Figure 7.** Monthly zonal mean frequency of occurrence of moist cirrus (left) and dry cirrus (right) over the SH (top) and NH (bottom). Shown with black contours are the monthly SPH variations relative to the annual mean SPH (left), and the monthly temperature variations relative to the annual mean temperature (right). Nonnegative SPH and temperature variations are shown with solid contours, and negative SPH and temperature variations are shown with solid contours.

clouds are indeed coupled to tropical convection. The seasonal variations of  moist cirrus are thus controlled by the seasonally varying Hadley cells, the intertropical convergence zones (ITCZ), and monsoons. The maximum frequency of occurrence of  moist cirrus clouds occur at around  11 °N–12 °N in the boreal summer and  11 °S–12 °S in the austral summer, with the boreal summer maximum larger than the austral summer maximum. The asymmetry between the NH and SH maxima is associated with the asymmetry of the ITCZ, which arises from the different shapes of the continents in the NH and SH [*Xie*, 2004].

 While the pattern of moist cirrus occurrence migrates between the NH and SH seasonally, the pattern of dry cirrus occurrence does not move significantly in the meridional direction with seasons. Dry cirrus obtain a single but broad maximum frequency of occurrence in the boreal  winter–spring within 10° of the equator (see  Figs. 6 and 8b).  Figure 8(b) further shows that the seasonal pattern of

[Figure]

**Figure 8.** Monthly–meridional distributions of the vertical maximum, zonal mean frequency of cirrus occurrence (top), and the zonal mean ice water path (IWP in g m$^{-2}$, bottom). The left panels show  moist cirrus with the zonal mean precipitation (mm d$^{-1}$) in black contours, and the right panels show  dry cirrus with the zonal mean CPT temperature (K) in black contours.

 the vertical maximum of dry cirrus occurrence is negatively correlated with the seasonal pattern of the CPT temperature. The vertical maximum of the frequency of occurrence of dry cirrus is located at around 16 km (recall Fig. 1b), so Fig. 8(b) generally captures the behavior of dry cirrus at high altitudes in the TTL and lower stratosphere. The seasonal cycle of the CPT is driven by the seasonal cycle of stratospheric planetary waves in the extratropical latitudes [*Yulaeva et al.*, 1994;

310   *Highwood and Hoskins*, 1998; *Jucker and Gerber*, 2017]. During the boreal winter, stronger wave activities in the extratropics result in stronger upwelling of the BDC and lower CPT temperatures [*Yulaeva et al.*, 1994; *Holton et al.*, 1995; *Highwood and Hoskins*, 1998]. In the cold TTL during the boreal winter, local negative temperature perturbations such as those generated by waves can readily increase the RH above the threshold for ice nucleation and so clouds are formed frequently.  The maximum

315   center of the upwelling of the BDC is located in the SH in the boreal winter [*Mote et al.*, 1996; *Plumb and Eluszkiewicz*, 1999], so the maximum frequency of occurrence of dry cirrus also occurs in the SH in the boreal winter.

The seasonal cycles of  moist and high-altitude dry cirrus described above are generally consistent with previous studies of cirrus clouds below 14 km–15 km [*Sassen et al.*, 2008; *Virts and Wallace*, 2010; *Nee and Lu*, 2021] and cirrus clouds above 14 km–15 km [*Tseng and Fu*, 2017; *Nee and Lu*, 2021], respectively.

320 However, the significance here is that we are able to distinguish  moist and dry cirrus from each other despite the overlapping in their vertical distributions (see Fig. 1b). By separating  moist and dry cirrus from each other, we can clearly demonstrate the relationships between  moist cirrus and convection, and between  high-altitude dry cirrus and the temperature in the TTL and  lower stratosphere. In previous studies, a particular altitude level (typically around 14 km–15 km) was chosen as a threshold to sep-

325 arate low- and high-altitude cirrus. In addition, much of previous research on the effect of temperature anomalies on cirrus clouds has focused on the higher altitudes in and above the TTL only. Here, the vertical profile of dry cirrus (see Fig. 1b) reveals a population of dry cirrus driven by negative temperature anomalies in the troposphere. We will return to discuss these low-altitude dry cirrus clouds in the troposphere in Section 4.3 below.

**4.3 Ice water contents**

330 Figure 9 shows the distributions of the occurrence of  moist and dry cirrus against temperature and in-cloud IWC. The frequency of occurrence of  moist cirrus is maximum in the temperature range between  210 K and 250 K, with IWCs between  $10^{-2}$ g m$^{-3}$ and $10^{-1}$ g m$^{-3}$. In comparison, the histogram of  dry cirrus shows a distinct maximum count between 190 K and 200 K, which is around the CPT temperature. The IWC of the peak distribution of  dry cirrus is between  $10^{-4}$ g m$^{-3}$ and $10^{-3}$ g m$^{-3}$. However,

335  dry cirrus are also occasionally detected below the TTL at temperatures above 200 K (altitudes below 14 km, see also Fig. 1b). These  dry cirrus at low altitudes have IWCs comparable to those of  moist cirrus at the same temperature/altitude levels.

Figure 9 further shows that the IWC in cirrus clouds increases with increasing temperature (decreasing altitude), which is consistent with previous observations [*Schiller et al.*, 2008; *Krämer et al.*, 2016; *Heymsfield et al.*, 2017; *Krämer et al.*, 2020].

340 The behavior of the in-cloud IWC as an increasing function of temperature holds regardless whether the clouds are  moist or dry cirrus. Thus,  dry cirrus typically contain less ice water than  moist cirrus because dry cirrus typically form at lower temperatures (higher altitudes). The different formation mechanisms (convection or non-convective processes) govern the temperature range in which cirrus clouds are formed, through which they govern the IWC  in the clouds.

345 As a consequence of the in-cloud IWC being a function of temperature regardless of cloud types, the vertical profiles of the in-cloud IWC of moist and dry cirrus are very similar (see the dashed lines in Fig. 10).

 On the other hand, the grid-average IWC due to dry cirrus is about an order of magnitude less than that  due to moist cirrus throughout most of the atmosphere except above about 15.5 km  (see the solid lines in Fig. 10). The grid-average

350 IWC depends on both the in-cloud IWC and the frequency of occurrence of clouds. Given that the IWCs in

[Figure]

**Figure 9.** Histogram of cloud samples against temperature and in-cloud IWC for (a)  moist cirrus and (b)  dry cirrus.

[Figure]

**Figure 10.** Vertical profiles of the  climatological tropical  mean grid-average IWC (solid) and  in-cloud IWC (dashed) associated with moist and dry cirrus.

 moist and dry cirrus are comparable  to each other at each temperature/altitude level, the difference in the  grid-average IWC between moist and dry cirrus is determined mainly by the difference in the frequency of occurrence between the two types of clouds. The 15.5 km level marks the altitude above which the frequency of occurrence (see Fig. 1b) and therefore the  grid-average IWC of dry cirrus exceed those of  moist cirrus.

Finally, Figs. 8(c) and (d) show the seasonal cycle of the ice water path (IWP)  due to moist and dry cirrus. The IWP is dominated by the ice water at low altitudes (see Fig. 10). Therefore, the seasonal cycle of the IWP reflects the seasonal cycle of cirrus clouds at low altitudes. For  moist cirrus, the seasonal patterns of the IWP and the maximum frequency of occurrence located at 14 km (see Fig. 1b) are similar. This indicates that  moist cirrus throughout the troposphere are coupled to each other and to convection. On the other hand,  by comparing Figs. 8(b) and (d), we can see that, for dry cirrus, the seasonal pattern of the IWP is different from that of the  vertical maximum of the frequency of occurrence (which is located  near 16 km in the TTL, see Fig. 1b).  While high-altitude dry cirrus are mostly located in the deep tropics, low-altitude dry cirrus occur over all latitudes of the tropics, from the equatorial region to the northern and southern edges of the tropics. These behaviors suggest that dry cirrus at low altitudes are decoupled from dry cirrus at high altitudes. We posit that low-altitude dry cirrus are driven by gravity  waves' temperature perturbations in the troposphere. Gravity wave activities are expected to be more prevalent in the winter months than the summer months in each hemisphere, consistently with the seasonal pattern of the IWP of dry cirrus clouds in Fig. 8(d).

**5 Summary**

Based on the monthly anomalies of the SPH and temperature in cloudy air relative to  all-sky condition, we have separated the population of tropical cirrus clouds detected by CALIPSO into ~~those of convective origin (convective cirrus ) and those of non-convective origins (non-convective cirrus). Convective cirrus occur in moist conditions and include (i) those that form from the freezing of liquid cloud droplets in convective updrafts, (ii) those that form by in situ ice nucleation from the vapor phase due to the adiabatic or diabatic cooling at the top of deep convection, and (iii) those that form by in situ ice nucleation in the moist air of the convective outflows. Non-convectivedry conditions and form by in situ ice nucleation in response to negative temperature anomalies~~ air that is colder and contains less moisture than usual.

 Moist cirrus are on average located at lower altitudes than  dry cirrus. The level of maximum  moist cirrus occurrence is  near 14 km, i.e., the bottom of the TTL. In comparison,  dry cirrus obtain their maximum frequency of occurrence at 16 km. The ratio of the number of  moist cirrus to the number of  dry cirrus is on the order of 2:1 over all altitudes of the tropical atmosphere, 3:1

below 14 km, 1:1 above 14 km, and 1:3 above the CPT. The majority of  dry cirrus are located above 14 km, but there are also  dry cirrus below 14 km.  Dry cirrus at high altitudes occur  in the deep tropics, while dry cirrus at low altitudes occur  all over the tropics, from the equatorial region to the northern and southern edges of the tropics.

The seasonal cycle of  moist cirrus is consistent with that of tropical convection, while the seasonal cycle of  dry cirrus above 14 km is consistent with that of the CPT. There are two maxima in the frequency of occurrence of  moist cirrus, one at around  11 °S–12 °S in the austral summer, and the other at around  11 °N–12 °N in the boreal summer. In contrast,  dry cirrus above 14 km occur most frequently  in the deep tropics in the boreal winter when the CPT is coldest.  Dry cirrus below 14 km occur most frequently in the winter months of each hemisphere whence wave activities are strongest.  These results suggest that the monthly occurrence of moist cirrus, high-altitude dry cirrus, and low-altitude dry cirrus are driven by different process mechanisms: (i) the occurrence of moist cirrus is driven by the moistening effect of convection, (ii) the occurrence of high-altitude dry cirrus is driven by the adiabatic cooling associated with the BDC as well as wave activities in the TTL and lower stratosphere, and (iii) the occurrence of low-altitude dry cirrus is driven by wave activities in the troposphere.

The IWC in both  moist and dry cirrus increases with increasing temperature (decreasing altitude). Thus,  dry cirrus—which on average occur at lower temperatures (higher altitudes)—tend to have lower IWCs than  moist cirrus. However, at a given altitude, the IWCs in  moist and dry cirrus are comparable to one another . Fresh outflow convective anvil cirrus may have much larger IWCs, but subsequent processes during their life cycles such as ice sedimentation and sublimation, and cloud horizontal spreading  can decrease the IWCs by several orders of magnitude as shown in previous  studies [*Boehm et* al., 1999; *Luo and Rossow*, 2004; *Dinh et al.*, 2010, 2012; *Gehlot and Quaas*, 2012; *Dinh et* al., 2014; *Jensen et* al., 2018; *Gasparini et* al., 2021].

The method proposed here to study cirrus clouds can be applied in model development to improve the representation of cirrus clouds in numerical simulations. We have demonstrated that the spatiotemporal distribution of cirrus clouds is governed by the SPH, temperature, and their variations. Therefore, models would need to accurately represent the SPH, temperature, and their variances in order to accurately simulate the distribution of cirrus clouds. It would be useful to compare between observations and numerical simulations in terms of the frequency and magnitude of the moisture and temperature anomalies and how they affect the occurrence of cirrus clouds. Such a comparison would reveal the specific strategies on how to adjust the model parameterization schemes (e.g., the convection scheme, the gravity wave drag scheme, and/or the microphysics scheme) to improve the representation of cirrus clouds in models.

*Author contributions.* Qin Huang carried out the analysis of the data. Both Qin Huang and Tra Dinh contributed to writing the manuscript.

*Competing interests.* No competing interests are present.

*Acknowledgements.* We thank D. Winker for the discussions at the AGU Fall Meeting in 2019, which motivate this work. We thank Blaž Gasparini and an anonymous reviewer for their feedback and comments which help improving the manuscript.

---

## Author Comment (AC2)

**Response to Reviewer 2**

**1 Overall assessment**

***Reviewer*** — The authors use CALIPSO measurements of tropical cirrus, along with ERA5 reanalysis temperature and specific humidity fields, to categorize cirrus as either convective or non-convective. They define convective clouds as those for which the specific humidity is greater than the annual mean at the location where the clouds were observed. As described below, I am skeptical that the paper adds any new understanding of the roles of convection and non-convective processes in governing the occurrence of cirrus clouds, which is the stated goal of the paper. Further, I believe the authors' definition of convective versus non-convective cirrus is misleading because it includes cirrus formed in situ in air masses with relatively high specific humidity that might (or might not) have been caused by convection somewhere upstream of the observed cirrus.

***Authors*** — We thank the reviewer for their insightful and constructive feedback. Prompted by the feedback from both reviewers, we have made changes to the manuscript and believe it has been significantly improved as a result. The list of major changes to the manuscript is attached. It may be helpful for the readers to refer to the list of changes before reading the discussions below. We are also attaching two versions of the revised manuscript, one with tracked changes and one without tracked changes. The discussions below refer to the line numbers in the version with tracked changes.

In response to the question whether "the paper adds any new understanding of the roles of convection and non-convective processes in governing the occurrence of cirrus clouds", we have revised the manuscript to discuss the contribution of our work in comparison with existing knowledge. In particular, the following text is added to lines 99–111 in the introduction:

"The knowledge that tropical cirrus clouds are driven by convection as well as by negative temperature anomalies associated with non-convective processes is not new (see the references cited above). However, to the best of our knowledge, this is the first time that individual vertical profiles of moist cirrus (that are related to convection) and dry cirrus (that are unrelated to convection) are obtained from observational data. In previous observational studies, a particular altitude (typically around 14 km–15 km, the bottom of the TTL) was chosen as a threshold level to separate cirrus driven by convection at lower altitudes and those which are driven by negative temperature anomalies at higher altitudes. Such a clear-cut separation may not be appropriate as the vertical profiles of moist and dry cirrus that we obtain here suggest that convection and non-convective sources contribute almost equally to the total occurrence of cirrus clouds above 14 km. Furthermore, the vertical profile of dry cirrus reveals that a significant number of cirrus clouds in the troposphere

are formed by negative temperature anomalies unrelated to convection. Much of previous research on the effect of temperature anomalies on cirrus clouds has focused on the higher altitudes in and above the TTL only. The other significant finding is that moist and dry cirrus contain similar IWCs if they are located at the same altitude or temperature level. Previously, cirrus that originate from convection were often thought to have larger IWCs (see e.g. Wang and Dessler, 2012)."

In response to the reviewer's comment that "the definition of convective versus non-convective cirrus is misleading", we have changed the terminology used in the manuscript, from cirrus that originate from convection and non-convective processes to moist and dry cirrus, respectively. The new names refer to how we classify the clouds based on the specific humidity (SPH) and temperature anomalies. The new terminology does not change the main results of the analysis, i.e., that the monthly spatiotemporal distribution of moist cirrus (formerly convective cirrus) is consistent with that of convection, while the monthly spatiotemporal distribution of dry cirrus (formerly non-convective cirrus) is distinct from that of convection.

Please find below our point-by-point reply, first to the major comments, followed by the minor comments of the reviewer.

**2  Major comments**

1. ***Reviewer*** — As noted by the authors, their definition of "convective-origin" cirrus includes clouds that form well downstream of the convection in air masses with relatively high specific humidity. This definition is much broader than the conventional definition of convective cirrus, which is limited to cirrus produced directly by the deep convection. Taken to the extreme, the authors' definition could include all cirrus in the tropical upper troposphere since deep convection is the primary source of upper tropospheric water vapor. The abstract states that convective cirrus are three times more common than non-convective cirrus. I fear readers will take this statement as the key message of the paper without understanding that the authors' definition of convective cirrus includes any clouds formed downstream of locations where deep convection hydrated the upper troposphere. I believe the authors should choose different terminology to avoid sending a misleading message.

   A related issue is that convection is not the only mechanism that can produce specific humidity at a particular location that is higher than the climatological mean. Transport of water from moist to dry regions can also produce a positive humidity anomaly. Further, the mean values used here are apparently annual means (the authors need to be more explicit on this point), in which case the specific humidity anomalies they are using could be dominated by seasonal variability. An alternate approach would be to use regional, seasonal (or subseasonal) mean temperature and humidity based on averages over some nearby domain and limited time period.

   ***Authors*** — As noted above, we have changed the terminology used in the manuscript, from cirrus that originate from convection and non-convective processes to moist and dry cirrus, respectively. The new terminology helps to avoid the confusion with the conventional definition of convective cirrus, which is often limited to cirrus produced directly by deep convection.

   The transport of water from moist to dry regions can produce positive SPH anomalies, but even in these cases the source of moisture must be the convective outflows that have been transported

horizontally by the winds (see Salathé and Hartmann, 1997, Sohn et al., 2008, Das et al., 2011). This point is discussed on lines 170–175 in the manuscript.

The mean values used are indeed the climatological annual means. We explicitly explained this point as: "The time-average SPH ($\overline{q}$) is obtained by averaging the SPH over all the times in the dataset regardless of whether clouds are present in the grid box." Please refer to lines 154–157. We are in fact targeting the monthly and seasonal variabilities in this work. Therefore, the usage of the annual means is appropriate. We have also changed the title of the manuscript to "Monthly occurrence of tropical cirrus clouds explained by monthly moisture and temperature variations" to clarify the focus on the monthly variabilities.

2. ***Reviewer*** — As noted above, the authors state in the abstract that most tropical cirrus originates from convection. However, their results show a strong height variation in the relative abundance of convective versus non-convective cirrus, with the former dominating below about 15 km, and the latter dominating in the upper TTL. This result is consistent with other data analysis and modeling studies. For example, Jensen et al. (2017) and Schoeberl et al. (2019) used satellite and airborne measurements of clouds and humidity to show that temperature effects dominate in the upper TTL, whereas deep convection controls clouds and humidity in the lower TTL. A number of modeling studies have shown that the observed occurrence frequency and regional distribution of TTL cirrus can be reproduced with only in situ cloud formation (e.g., Ueyama et al., 2015; Ueyama et al., 2018; Schoeberl et al., 2019). Therefore, the conclusion that convection dominates clouds and humidity in the lower TTL, whereas temperature variability dominates cirrus formation in the upper TTL is certainly not new.

As noted by the authors, a number of past studies have shown that TTL cirrus tend to form in anomalously cold air masses. Therefore, the authors' finding that most cirrus in the TTL occur in times and locations where the temperature is below the climatological mean is no surprise. In general, I do not think the analysis here really clarifies the relationship between convection and tropical cirrus beyond what was already known.

***Authors*** — We have discussed the contribution of our work in comparison with existing knowledge in the revised manuscript. Please see lines 99–111 and also our response to the reviewer's overall assessment in Section 1 above. The significance of our work is that this is the first time that individual vertical profiles of moist cirrus (that are related to convection) and dry cirrus (that are unrelated to convection) are obtained from observational data. The vertical profiles of moist and dry cirrus are obtained in the troposphere as well as the TTL and the lower stratosphere. The vertical profile of dry cirrus reveals that a significant number of cirrus clouds in the troposphere are formed by negative temperature anomalies unrelated to convection. Much of previous research on the effect of temperature anomalies on cirrus clouds has focused on the higher altitudes in and above the TTL only.

The reviewer did not provide the full reference to the paper by Jensen et al. (2017). We assume that the reviewer referred to the study on the relative humidity (RH) over the Pacific by Jensen et al. (2017). Jensen et al. (2017) studied the influences of convection and temperature anomalies on the RH while we study their influences on the occurrence of cirrus clouds. We did not find the vertical profiles of cirrus clouds from convective and non-convective sources in Jensen et al. (2017). Furthermore, the data used by Jensen et al. (2017) is limited to the Pacific in the ATTREX field campaigns in the boreal winter in 2013 and 2014. On the other hand, our analysis is carried out for the entire tropics over the 9-year period from January 2007 to December 2015.

The reviewer did not provide the full reference to the paper by Schoeberl et al., 2019, either. We assume that the reviewer referred to the study on water vapor and clouds in the TTL by Schoeberl et al. (2019). Figure 10 in Schoeberl et al. (2019) shows the vertical profiles of cloud fraction in the TTL simulated in a one-dimensional model under the influences of convection and gravity waves. In contrast, our results are obtained based on observational data. Our results are useful as they can be compared with modelling results such as those in Schoeberl et al. (2019) in order to improve both the understanding of cirrus clouds and the accuracy of model simulations of cirrus.

The studies by Ueyama et al. (2015, 2018) are modelling studies, in contrast to our work which is an observational study. The findings from these modelling studies are important in their own rights and these two papers are cited in the manuscript on lines 71–75.

Indeed, as pointed out by the reviewer, "a number of past studies have shown that TTL cirrus tend to form in anomalously cold air masses". The studies by Jensen et al. (2017), Schoeberl et al. (2019), Ueyama et al. (2015, 2018) mentioned above by the reviewer are also limited to the TTL only. We contribute to the existing literature by showing that a significant number of cirrus clouds in the troposphere are also formed by negative temperature anomalies unrelated to convection.

**3   Minor comments**

1. ***Reviewer*** — Abstract, lines 8–9: The authors state that "The remaining clouds that are not directly influenced by convection are driven by negative temperature perturbations." As shown later in the paper, this statement is not correct. Some of the clouds with negative SPH perturbations have positive temperature perturbations. In fact, this "unidentified" category comprises 10 % of the cloud samples, which is not negligible.

   ***Authors*** — We have revised the abstract and corrected this mistake. The sentence "The remaining clouds that are not directly influenced by convection are driven by negative temperature perturbations." has been deleted. Please see lines 11–12.

2. ***Reviewer*** — Line 32: The authors should cite Forster and Shine (2002) since this paper quantified the impact of stratospheric humidity on the radiation budget well before the papers cited.

   ***Authors*** — Thank you for pointing out this relevant paper. We have now cited Forster and Shine (2002) on line 38.

3. ***Reviewer*** — Line 48: When discussing dissipation of convective cirrus, the authors mention cloud horizontal spreading and sublimation, and they cite a series of Dinh et al. papers. However, these papers did not address the evolution of convective cirrus; further, other studies have shown that sedimentation is the dominant process reducing the IWC of convective cirrus Boehm et al. (1999); Jensen et al.(2018).

   ***Authors*** — We have corrected the sentence to include sedimentation as a process that governs the dissipation of cirrus. The papers by Boehm et al. (1999), Jensen et al. (2018), as well as Luo and Rossow (2004), Gehlot and Quaas (2012), Gasparini et al. (2021), in which the evolution of convective cirrus was addressed, are now cited here. Please see lines 54–57.

4. ***Reviewer*** — Lines 60–61: The authors state that "Ueyama et al. (2015) calculated backward trajectories that end at the tropopause..." The study actually used curtains along trajectories to simulate TTL cirrus. Further, the authors state that Luo and Rossow (2004) "simulated clouds along the trajectories." As far as I can tell, this paper did not use cloud simulations. They simply tracked the clouds in the satellite imagery.

   ***Authors*** — We have revised the text to describe the work by Ueyama et al. (2015, 2018) more appropriately. Please see lines 71–75. Thank you for pointing out the mistake with the referencing of the Luo and Rossow (2004) paper. The sentence discussing that Luo and Rossow (2004) "simulated clouds along the trajectories" has been deleted. The paper by Luo and Rossow (2004) is no longer cited here but is cited on line 56.

5. ***Reviewer*** — Lines 68–69: In addition to the papers cited here, Jensen and et al. (2020) recently documented cases of convective hydration of the lower stratosphere.

   ***Authors*** — The paper by Jensen et al. (2020) is now cited here. Please see line 89.

6. ***Reviewer*** — Lines 117–121: The mean specific humidity and temperature are apparently averaged over the entire data time period (although this is not stated explicitly). This would mean the averages are climatological, annual means, and the resulting correlations between cloud occurrence, humidity anomalies, and temperature anomalies will largely just represent the seasonal variations in these quantities. I think it would make more sense to use seasonal means for the clear-sky averages.

   ***Authors*** — The mean SPH and temperature are indeed averaged over the entire data period. We have explained this explicitly on lines 156–157 in the manuscript and also in our response to the reviewer's major comment number 1 above. Our work specifically addresses the seasonal variations in clouds, SPH, and temperature. For this purpose, we think that it is appropriate to use the annual means for the all-sky fields.

7. ***Reviewer*** — Lines 171–177: The authors note that the peak of the non-convective cloud frequency occurs below the CPT, and as an explanation, they cite the modeling study indicating that radiatively-driven circulations will be damped if the temperature in the cloud increases with height. First, there is no observational evidence showing that radiatively-driven circulations routinely occur in cirrus near the tropopause, and the lifetime of TTL cirrus (limited by wave-driven temperature perturbations) is typically too short for these circulations to develop (Jensen et al., 2011). Second, there is a much simpler explanation for the peak cloud frequency altitude occurring below the tropical mean CPT: The CPT altitude varies considerably from location to location and time to time. Therefore, at the mean CPT altitude, you are often above the local CPT, in which case cloud formation would be suppressed. As shown by Pan and Munchak (2011), TTL cirrus cloud tops generally occur just below the local CPT.

   ***Authors*** — Indeed, the role of the radiatively-driven circulation in the life cycles of cirrus clouds has been shown in numerical simulations only. Observational evidence is not yet available. We have revised the sentence to stress that this is a modelling result, not observational result. Please see line 219.

   We agree with the reviewer that, above the local CPT, the RH is low so the formation of clouds is suppressed. In the manuscript, we have included this point as the first potential reason why the frequency of occurrence of cirrus clouds peaks below the CPT altitude. Please see lines 217–219.

8. ***Reviewer*** — Lines 185–192: The authors use their convective occurrence height distribution to estimate the frequency of overshooting above the CPT. I do not believe there is any quantitative value to this calculation. First, it is entirely possible that the few convective clouds (with positive SPH anomalies) observed above the CPT are just places where the SPH is above the climatological mean with no relationship to deep convection. Second, as noted above, the CPT height varies spatially and temporally. Therefore, for this calculation to have any value, the authors would need to carefully determine which of the clouds are above the local CPT.

   ***Authors*** — We have removed these lines from the manuscript (lines 236–242).


90  suggest that moist cirrus are driven by convection. In contrast, the spatiotemporal distribution of dry cirrus is distinct from that of tropical convection, suggesting that these clouds are unrelated to convection. The seasonal cycle of

95  dry cirrus in and above the TTL is consistent with that of  the cold point tropopause (CPT) temperature, and the seasonal cycle of  dry cirrus below the TTL is consistent with that of  wave activities in the troposphere.

100 The knowledge that tropical cirrus clouds are driven by convection as well as by negative temperature anomalies associated with non-convective processes is not new (see the references cited above). However, to the best of our knowledge, this is the first time that individual vertical profiles of moist cirrus (that are related to convection) and dry cirrus (that are unrelated to convection) are obtained from observational data. In previous observational studies, a particular altitude (typically around 14 km–15 km, the bottom of the TTL) was chosen as a threshold level to separate cirrus driven by convection at lower altitudes and those which are driven by negative temperature anomalies at higher altitudes. Such a clear-cut separation may not be

105 appropriate as the vertical profiles of moist and dry cirrus that we obtain here suggest that convection and non-convective sources contribute almost equally to the total occurrence of cirrus clouds above 14 km. Furthermore, the vertical profile of dry cirrus reveals that a significant number of cirrus clouds in the troposphere are formed by negative temperature anomalies unrelated to convection. Much of previous research on the effect of temperature anomalies on cirrus clouds has focused on the higher altitudes in and above the TTL only. The other significant finding is that moist and dry cirrus contain similar IWCs if

110 they are located at the same altitude or temperature level. Previously, cirrus that originate from convection were often thought to have larger IWCs [see e.g. *Wang and Dessler*, 2012].

The remaining of the paper is organized as follows. The data and methodology are described respectively in Sections 2 and 3. Section 4 discusses the characteristics of the occurrence of  moist and dry cirrus in the tropics, including their spatiotemporal distributions and IWCs. Section 5 contains the summary.

**115  2   Data**

We analyze the monthly-mean, three-dimensional cirrus cloud occurrence and IWC of the Lidar Level 3 Ice Cloud Data, Standard Version 1-00 [*NASA Langley Atmospheric Science Data Center*, 2018]. This data was collected with the Cloud-Aerosol LIdar with Orthogonal Polarization (CALIOP) instrument on  CALIPSO [*Winker et al.*, 2010]. CALIOP is capable of detecting clouds with optical depths of 0.01

120 or less [*Winker et al.*, 2007]. For this reason, CALIOP data are well suited for studies of cirrus clouds, many of which are optically thin. The user guides for the monthly Lidar Level 3 Ice Cloud Data can be found online (see https://www-calipso.larc.nasa.gov/resources/calipso_users_guide/qs/cal_lid_l3_ice_cloud_v1-00.php and https://www-calipso.larc.nasa.gov/resources/calipso_users_guide/qs/cal_lid_l3_cloud_occurrence_v1-00.php).

The spatial resolution of the monthly Lidar Level 3 Ice Cloud Data is 2.5° in the zonal direction, 2.0° in the meridional

125 direction, and 120 m in the vertical. The monthly data is not suitable to conduct case study of clouds that occur in response to individual convection or wave events. However, if convection and wave activities have intrinsic seasonal cycles, their effects on the seasonal cycles of cirrus occurrence should be captured in the monthly data. Thus, this data is appropriate for us to study the

[Figure]

**Figure 1.** Climatological mean frequency of occurrence of cirrus clouds in the tropics: (a) latitude–altitude profile of the zonal mean frequency  and (b) vertical profile of the frequency of occurrence averaged over the tropics.

spatial distribution of cirrus clouds on the seasonal and climatological time scales. Figure 1 shows the frequency of occurrence of cirrus clouds in the tropics between 24 °S and 24 °N over the 9-year period from January 2007 to December 2015 obtained from CALIPSO. The overall spatial distribution of cirrus clouds in the figure is consistent with previous observational studies using CALIPSO [*Sassen et al.*, 2008; *Mace et al.*, 2009; *Hong and Liu*, 2015] and satellite radiometers [*Wang et al.*, 1996].

 For the meteorological conditions surrounding cirrus clouds,  specifically the specific humidity (SPH) and temperature of the atmosphere, we use the ERA5 data [*Hersbach et* al., 2020]. In addition, we obtain the data for precipitation from the Version-2 Global Precipitation Climatology Project (GPCP) Monthly Precipitation Analysis [*Adler et al.*, 2003]. The  SPH, temperature, and precipitation data were downloaded for the same temporal and spatial domains as the ice cloud data above, and then they were interpolated to the same grid as the ice cloud data.

**3 Identifying  moist and dry cirrus**

Figure 2 shows the vertical profiles of the relative humidity (RH) that has been averaged over time and the tropical domain in cloudy and  all-sky conditions. The RH discussed in this work is specifically with respect to ice. At a given grid box location,  cloudy conditions refer to the times when cirrus clouds are detected, i.e., when the ice cloud fraction is greater than 0.01, in the grid box. The figure shows that on average the RH is greater in cloudy conditions than in  all-sky conditions at every altitude. This is consistent with existing observations [*Sandor et al.*, 2000; *Kahn et al.*, 2008, 2009; *Krämer et al.*, 2020] that the RH is greater in cloudy conditions to support the formation and maintenance of  clouds.

[Figure]

**Figure 2.** Vertical profiles of the climatological  mean RH in cloudy and  all-sky conditions.

The RH is related to the temperature ($T$) and the SPH ($q$) via

$$\text{RH} = \frac{p_{\text{v}}}{e_{\text{si}}(T)} = \frac{R_{\text{v}}}{R}\frac{q}{e_{\text{si}}(T)}p, \tag{1}$$

where $R = 287\,\text{J}\,\text{kg}^{-1}\,\text{K}^{-1}$ and $R_{\text{v}} = 461\,\text{J}\,\text{kg}^{-1}\,\text{K}^{-1}$ are respectively the specific gas constants of air and water vapor, $p$ is atmospheric pressure, $p_{\text{v}}$ is the partial pressure of water vapor in air, and $e_{\text{si}}(T)$ is the saturation water vapor pressure with
150  respect to ice, a function of temperature. The function $e_{\text{si}}(T)$ increases with temperature and is calculated based on the empirical formula given by *Murphy and Koop* [2005]. According to Eq. (1), at a given pressure level, large RH values inside clouds relative to  the all-sky condition must arise from positive SPH anomalies and/or negative temperature anomalies.

 Let $\Delta q(x,y,z,t) = q_{\text{cld}}(x,y,z,t) - \overline{q}(x,y,z)$ denote the difference between the
155  SPH in the cloud sample at the location $(x,y,z)$ and time $t$ and the  time-average SPH at the same location. The  time-average SPH ($\overline{q}$) is obtained by averaging the SPH over  all the times in the dataset regardless of whether clouds are present in the grid box. Similarly,  $\Delta T(x,y,z,t) = T_{\text{cld}}(x,y,z,t) - \overline{T}(x,y,z)$ is the difference between the temperature in the cloud sample at the location $(x,y,z)$ and time $t$ and the  time-average
160  temperature at the same location. The vertical profiles of the climatological mean, tropical average $\Delta q$ and $\Delta T$ are shown in Fig. 3. The figure shows that $\Delta q$ (green line) is positive in the troposphere, indicating that most cirrus clouds in the troposphere are formed and maintained in the months of positive SPH anomalies. Furthermore, $\Delta q$ decreases exponentially with altitude, consistent with the fact that the background SPH decreases exponentially with altitude. On the other hand, the magnitude of the temperature anomalies experienced by cirrus clouds is small in most of the troposphere, and it becomes significant only above
165  14 km or so. The result that cirrus clouds above 14 km experience significant negative temperature anomalies is consistent with previous studies [*Boehm and Verlinde*, 2000; *Virts et al.*, 2010; *Virts and Wallace*, 2010; *Tseng and Fu*, 2017].

[Figure]

**Figure 3.** Vertical profiles of the climatological mean, tropical average differences in (a) SPH and (b) temperature between cloudy and  all-sky conditions.

 Let us refer to the clouds in which $\Delta q > 0$ as moist cirrus. We expect that moist cirrus are influenced by convection since positive SPH anomalies are largely produced by the upward transport of moisture by convection in the free atmosphere in the tropics. Some positive SPH anomalies may be located away from active convection, but even in these cases the source of moisture must be the convective outflows that have been transported horizontally by the winds [see *Salathé and Hartmann*, 1997; *Sohn et al.*, 2008; *Das et al.*, 2011].  Moist cirrus include clouds that form within the convective updrafts and at the top of convection, as well as those that form in the moist air of the convective outflows downstream of convection. Examples of the latter type of  moist cirrus were recently reported by *Cairo et al.* [2021]. Using this  definition, we find that  58 % of tropical cirrus clouds are  moist cirrus (see Fig. 4). About  60 % of moist cirrus experience positive temperature anomalies, and the  remaining 40 % experience negative temperature anomalies.

The remaining cirrus clouds in which $\Delta q \leq 0$ consist of two categories. The first category comprises of the cloud samples in which $\Delta q \leq 0$ and $\Delta T < 0$.  We refer to these clouds as dry cirrus. Dry cirrus make up 34 % of tropical cirrus clouds (see Fig. 4).  Since convection sometimes leads to dehydration of the air [*Jensen et* al., 2007], in principle some dry cirrus can be associated with convection. However, our analysis (see Section 4.2) shows that the monthly spatiotemporal distribution of dry cirrus is distinct from those of convection and moist cirrus. The maximum frequency of occurrence of dry cirrus is located remotely away from the maximum frequency of occurrence of moist cirrus. Thus, the negative temperature anomalies that form and maintain  dry cirrus are unlikely to be the cooling at the top of convection. Rather, they are associated with waves and/or the adiabatic cooling associated with the upwelling of the BDC.

[Figure]

**Figure 4.** Histogram of cloud samples against the differences in SPH and temperature between cloudy and all-sky conditions.

 The last category of clouds is for those in which $\Delta q \leq 0$ and $\Delta T \geq 0$. These clouds are  driven by neither positive SPH anomalies nor negative temperature anomalies. They are most likely in the decaying stage of their lifetimes. In these cases, the SPH and temperature anomalies cannot be used to identify their formation and maintenance mechanisms.

 Decaying clouds make up 8 % of tropical cirrus ~~clouds, the method described above allows us to identify the majority of cirrus clouds and their relationship with convection.Furthermore, as shown in Section 4 below, the spatial distributions and seasonal cycles of convective and non-convective cirrus are distinct from each other. The occurrence of convective cirrus is consistent with the location and the seasonal cycle of tropical convection, while the occurrence of non-convective cirrus is consistent with the CPT. The seasonal cycle of the CPT is strongly coupled to that of the BDC *Highwood and Hoskins*, 1998; *Jucker and Gerber*, 2017. These results suggest that the method we propose is appropriate to separate convective and non-convective cirrus~~(see Fig. 4).

**4 Characteristics of the occurrence of  moist and  dry cirrus**

**4.1 Spatial distributions**

Figures 1 and 5  show that the frequency of occurrence of  moist cirrus is maximum  near 14 km ($\sim$ 150 hPa), coincided with the level of zero net radiative heating rate, which is often defined as the bottom of the TTL [*Fueglistaler et al.*, 2009]. The 14 km altitude is also approximately the level of neutral buoyancy, which provides the upper bound for convective development in the vertical [*Takahashi and Luo*, 2012]. The level of maximum convective mass outflow is located several kilometers lower at around 10 km–11 km [*Takahashi and Luo*, 2012].  Moist cirrus between the level of neutral buoyancy (14 km) and the level of maximum convective outflow (10 km–11 km) are likely anvil cirrus.  Moist cirrus above 14 km are likely to originate from (i) the further lofting, spreading and detachment of anvils, (ii) in situ ice

[Figure]

**Figure 5.** (a) Latitude–altitude profiles of the climatological zonal mean frequency of occurrence filled colored contours) and  dry cirrus (white contours), and (b) vertical profiles of  the climatological zonal mean frequency of occurrence of moist and dry cirrus averaged over the SH (dashed) and  NH (solid).

nucleation in the moist air of the convective outflows in response to cold anomalies (see Fig. 3) associated with the cooling at the top of deep convection and/or waves. At lower altitudes (below 10 km or at temperatures above 235 K),  moist cirrus originate from mixed-phase clouds [*Heymsfield et al.*, 2017], i.e. they are of liquid origin [terminology following *Krämer et al.*, 2016].  More moist cirrus experience positive temperature anomalies than negative temperature anomalies (see Figs. 3 b and 4). The positive temperature anomalies likely arise from the latent heat release in convection.

 Dry cirrus tend to occur at higher altitudes than  moist cirrus. The frequency of occurrence of  dry cirrus maximizes at around 16 km, below the CPT (see Figs. 1 and 5). The climatological tropical mean CPT is found to be at 16.8 km. The level of maximum cirrus occurrence is capped above by the CPT potentially because of two reasons. Firstly, the RH decreases with altitude above the CPT as temperature increases with altitude (see Fig. 2). Thus, above the CPT the negative temperature perturbations must be of large magnitudes to raise the RH above the threshold of ice nucleation. Secondly, the modelling study by *Dinh et* al. [2010] suggested that a necessary condition for cirrus clouds to self-maintain for a long time is that the temperature in the cloud layer decreases with altitude. In this situation, the circulation induced by the cloud radiative heating produces in-cloud water vapor flux convergence that acts against ice sublimation [*Dinh et al.*, 2010]. On the other hand, when the temperature in the cloud layer increases with altitude (such as above the CPT), the circulation induced by the cloud radiative heating produces in-cloud water vapor flux divergence that enhances ice sublimation.  As a result, clouds above the CPT are short-lived and  so the frequency of cloud occurrence above the CPT is small.

**Table 1.** Percentage contributions of the different types of cirrus clouds to the total cirrus occurrence in different layers of the tropical atmosphere. The bottom of the tropical tropopause layer (TTL) is located near 14.0 km ($\sim$ 150 hPa). The climatological tropical mean CPT is at 16.8 km ($\sim$ 100 hPa).

| |  Moist |  Dry |  Decaying |
|---|---|---|---|
| Above CPT |  22 | 76 |  2 |
| Above 16.8 km | 25 | 73 | 2 |
| Above 14.0 km |  48 |  46 | 6 |
| Below 14.0 km | 67 | 26 | 7 |
| All altitudes |  58 |  34 |  8 |

Table 1 shows the percentage contributions of the different types of cirrus clouds to the total cirrus in different layers of the tropical atmosphere. The table shows that  moist cirrus dominate the entire atmosphere and the troposphere below 14 km, i.e., the bottom of the TTL. Above 14 km,  moist and dry cirrus contribute almost equally to the total cirrus cloud occurrence . This is consistent with the observational study by *Massie et al.* [2002] that convection affects half of cirrus population in the TTL, although they studied cirrus clouds over the maritime continent only. This result is also consistent with the modeling study by *Schoeberl* et al. [2018] in which cirrus cloud fraction in the TTL doubles when convection is included in the model simulations. Above the CPT,  dry cirrus make up the large majority (76 %) of clouds, but the percentage of  moist cirrus is not negligible ( 22 %). Similar numbers (73 % dry cirrus and 25 % moist cirrus) are obtained above 16.8 km, which is the altitude of the climatological tropical mean CPT .

~~Based on the vertical profile of the frequency of occurrence of convective cirrus clouds (see Fig. 1b) , we can estimate the degree of overshooting convection above the CPT. Above the CPT, the frequency of occurrence of convective cirrus decreases with altitude, indicating that the degree of penetration of convection into the stratosphere decreases with altitude. At the CPT, the frequency of occurrence of convective cirrus is 1.7 %. This provides the upper bound for the occurrence of overshooting convection injecting ice into the stratosphere because not all convective cirrus are formed within the convective updrafts; some convective cirrus are formed in situ in the moist air of the convective ouflows. Gettelman et al. [2002] found based on cloud brightness temperatures that convection is present above the CPT about 0.5 % of~~

Figure 5 shows that there are more moist cirrus clouds in the Northern Hemisphere (NH) than in the Southern Hemisphere (SH) at every altitude. This is consistent with the fact that convection is tronger in the NH than in the SH. On the other hand, above 15 km there are less dry cirrus clouds in the NH than in the SH, and below 15 km there are more dry cirrus clouds in the NH than in the SH. Above 15 km, the occurrence of dry cirrus is anti-correlated with the CPT temperature, which is

[Figure]

**Figure 6.** Latitude–altitude profiles of the zonal mean frequency of occurrence of moist cirrus (filled colored contours) and dry cirrus (white contours) in December–January–February (DJF), March–April–May (MAM), June–July–August (JJA), and September–October–November (SON).

The meridional pattern of convective cirrus occurrence is bimodal and asymmetric about the equator (Figs. 5a and **??**). There are two maxima at approximately 10 °S and 10 °N, with the northern hemisphere (NH ) maximum being larger than the southern hemisphere (SH) maximum, consistent with the fact that convection is stronger in the NH. In comparison, the meridional pattern of non-convective cirrus occurrence is unimodal, with the maximum frequency of occurrence centered around the equator (Figs. 5b and **??**). The different spatial distributions of convective and non-convective cirrus suggest that the mechanisms governing the occurrence of non-convective cirrus is distinct from convection . This topic is further discussed in Section 4.2.

Figures 5(c) and (d) show the grid-average ice mass density associated with convective and non-convective cirrus. For both types of clouds controlled by the upwelling of the BDC. The maximum center of the upwelling of the BDC is located in the SH [*Mote et* al., 1996; *Plumb and Eluszkiewicz, 1999*], so there are more dry cirrus in the SH above 15 km. To explain why below 15 km there are more dry cirrus clouds in the NH than the SH, the maximum ice mass density is located below we refer to Fig. 6

which shows the occurrence of moist and dry cirrus over the four seasons. In the  NH, dry cirrus below 15 km occur most frequently in the boreal winter (December–January–February) and to a lesser extent in the boreal spring (March–April–May). In these months of the year, convection is the least active in the NH and the most active in the SH, consistently with the distribution of moist cirrus clouds shown in Fig. 6. In other words, dry cirrus below 15 km are located remotely away (in the opposite hemisphere) from the most active convection. It is thus unlikely that dry cirrus clouds below 15 km are driven by convection, even though convection can produce negative temperature anomalies. We posit that these low-altitude dry cirrus are instead driven by the temperature anomalies associated with gravity waves in the troposphere. Observations of gravity waves in the tropical troposphere are rare. However, the particular radiosonde data over the tropical regions of the USA [*Zhang et* al., 2010] indeed show that the energy density of tropospheric gravity waves is maximum in the boreal winter–spring, consistent with the occurrence of low-altitude dry cirrus in the NH in these seasons.

**4.2 Seasonal cycles**

Figure 7 shows how the seasonal cycle of  moist cirrus is forced from the bottom up by the seasonal cycle of the SPH in the troposphere, while the seasonal cycle of  dry cirrus in the TTL is forced from the top down by the seasonal cycle of the temperature in the TTL and lower stratosphere. In each hemisphere,  moist cirrus occur most  frequently during the summer  months when the SPH perturbations are positive and least frequently during the winter months when the SPH perturbations are negative. The seasonal cycle of  moist cirrus in the NH is thus opposite of that in the SH, i.e., when the frequency of occurrence of  moist cirrus is maximum in the NH, it is minimum in the SH.  In contrast, in both the NH and SH  dry cirrus occur most  frequently in the boreal winter  when the temperatures in the TTL are minimum  and least frequently in the boreal summer when the temperatures in the TTL are maximum.

~~Climatological monthly zonal mean frequency of occurrence of convective cirrus (left) and non-convective cirrus (right) over the SH (top) and NH (bottom). Shown with black contours are the monthly SPH perturbations relative to the annual mean SPH (left), and the monthly temperature perturbations relative to the annual mean temperature (right). Positive (negative) SPH and temperature perturbations are shown with solid (dashed) contours.~~

Figure 8(a) shows the seasonal migrations of  moist cirrus and precipitation between the NH in the boreal winter and the SH in the austral summer. The similar seasonal patterns of  moist cirrus and precipitation suggest that these

[Figure]

**Figure 7.** Monthly zonal mean frequency of occurrence of moist cirrus (left) and dry cirrus (right) over the SH (top) and NH (bottom). Shown with black contours are the monthly SPH variations relative to the annual mean SPH (left), and the monthly temperature variations relative to the annual mean temperature (right). Nonnegative SPH and temperature variations are shown with solid contours, and negative SPH and temperature variations are shown with solid contours.

clouds are indeed coupled to tropical convection. The seasonal variations of  moist cirrus are thus controlled by the seasonally varying Hadley cells, the intertropical convergence zones (ITCZ), and monsoons. The maximum frequency of occurrence of  moist cirrus clouds occur at around  11 °N–12 °N in the boreal summer and  11 °S–12 °S in the austral summer, with the boreal summer maximum larger than the austral summer maximum. The asymmetry between the NH and SH maxima is associated with the asymmetry of the ITCZ, which arises from the different shapes of the continents in the NH and SH [*Xie*, 2004]. ~~Overall, there are more convective cirrus in the NH (~60 %) than the SH (~40 %).~~

 While the pattern of moist cirrus occurrence migrates between the NH and SH seasonally, the pattern of dry cirrus occurrence does not move significantly in the meridional direction with seasons. Dry cirrus obtain a single but broad maximum frequency of occurrence in the boreal  winter–spring within 10° of the equator (see  Figs. 6 and 8b).  Figure 8(b) further shows that the seasonal pattern of

[Figure]

**Figure 8.** Monthly–meridional distributions of the vertical maximum, zonal mean frequency of cirrus occurrence (top), and the zonal mean ice water path (IWP in $g\,m^{-2}$, bottom). The left panels show  moist cirrus with the zonal mean precipitation ($mm\,d^{-1}$) in black contours, and the right panels show  dry cirrus with the zonal mean CPT temperature (K) in black contours.

 the vertical maximum of dry cirrus occurrence is negatively correlated with the seasonal pattern of the CPT temperature. The vertical maximum of the frequency of occurrence of dry cirrus is located at around 16 km (recall Fig. 1b), so Fig. 8(b) generally captures the behavior of dry cirrus at high altitudes in the TTL and lower stratosphere. The seasonal cycle of the CPT is driven by the seasonal cycle of stratospheric planetary waves in the extratropical latitudes [*Yulaeva et al.*, 1994;

310   *Highwood and Hoskins*, 1998; *Jucker and Gerber*, 2017]. During the boreal winter, stronger wave activities in the extratropics result in stronger upwelling of the BDC and lower CPT temperatures [*Yulaeva et al.*, 1994; *Holton et al.*, 1995; *Highwood and Hoskins*, 1998]. In the cold TTL during the boreal winter, local negative temperature perturbations such as those generated by waves can readily increase the RH above the threshold for ice nucleation and so clouds are formed frequently.  The maximum

315   center of the upwelling of the BDC is located in the SH in the boreal winter [*Mote et al.*, 1996; *Plumb and Eluszkiewicz*, 1999], so the maximum frequency of occurrence of dry cirrus also occurs in the SH in the boreal winter.

The seasonal cycles of  moist and high-altitude dry cirrus described above are generally consistent with previous studies of cirrus clouds below 14 km–15 km [*Sassen et al.*, 2008; *Virts and Wallace*, 2010; *Nee and Lu*, 2021] and cirrus clouds above 14 km–15 km [*Tseng and Fu*, 2017; *Nee and Lu*, 2021], respectively.

320 However, the significance here is that we are able to distinguish  moist and dry cirrus from each other despite the overlapping in their vertical distributions (see Fig. 1b). By separating  moist and dry cirrus from each other, we can clearly demonstrate the relationships between  moist cirrus and convection, and between  high-altitude dry cirrus and the temperature in the TTL and  lower stratosphere. In previous studies, a particular altitude level (typically around 14 km–15 km) was chosen as a threshold to sep-

325 arate low- and high-altitude cirrus. In addition, much of previous research on the effect of temperature anomalies on cirrus clouds has focused on the higher altitudes in and above the TTL only. Here, the vertical profile of dry cirrus (see Fig. 1b) reveals a population of dry cirrus driven by negative temperature anomalies in the troposphere. We will return to discuss these low-altitude dry cirrus clouds in the troposphere in Section 4.3 below.

**4.3 Ice water contents**

330 Figure 9 shows the distributions of the occurrence of  moist and dry cirrus against temperature and in-cloud IWC. The frequency of occurrence of  moist cirrus is maximum in the temperature range between  210 K and 250 K, with IWCs between  $10^{-2}$ g m$^{-3}$ and $10^{-1}$ g m$^{-3}$. In comparison, the histogram of  dry cirrus shows a distinct maximum count between 190 K and 200 K, which is around the CPT temperature. The IWC of the peak distribution of  dry cirrus is between  $10^{-4}$ g m$^{-3}$ and $10^{-3}$ g m$^{-3}$. However,

335  dry cirrus are also occasionally detected below the TTL at temperatures above 200 K (altitudes below 14 km, see also Fig. 1b). These  dry cirrus at low altitudes have IWCs comparable to those of  moist cirrus at the same temperature/altitude levels.

Figure 9 further shows that the IWC in cirrus clouds increases with increasing temperature (decreasing altitude), which is consistent with previous observations [*Schiller et al.*, 2008; *Krämer et al.*, 2016; *Heymsfield et al.*, 2017; *Krämer et al.*, 2020].

340 The behavior of the in-cloud IWC as an increasing function of temperature holds regardless whether the clouds are  moist or dry cirrus. Thus,  dry cirrus typically contain less ice water than  moist cirrus because dry cirrus typically form at lower temperatures (higher altitudes). The different formation mechanisms (convection or non-convective processes) govern the temperature range in which cirrus clouds are formed, through which they govern the IWC  in the clouds.

345 As a consequence of the in-cloud IWC being a function of temperature regardless of cloud types, the vertical profiles of the in-cloud IWC of moist and dry cirrus are very similar (see the dashed lines in Fig. 10).

 On the other hand, the grid-average IWC due to dry cirrus is about an order of magnitude less than that  due to moist cirrus throughout most of the atmosphere except above about 15.5 km  (see the solid lines in Fig. 10). The grid-average

350 IWC depends on both the in-cloud IWC and the frequency of occurrence of clouds. Given that the IWCs in

[Figure]

**Figure 9.** Histogram of cloud samples against temperature and in-cloud IWC for (a)  moist cirrus and (b)  dry cirrus.

[Figure]

**Figure 10.** Vertical profiles of the  climatological tropical  mean grid-average IWC (solid) and  in-cloud IWC (dashed) associated with moist and dry cirrus.

moist and dry cirrus are comparable to each other at each temperature/altitude level, the difference in the grid-average IWC between moist and dry cirrus is determined mainly by the difference in the frequency of occurrence between the two types of clouds. The 15.5 km level marks the altitude above which the frequency of occurrence (see Fig. 1b) and therefore the grid-average IWC of dry cirrus exceed those of moist cirrus.

Finally, Figs. 8(c) and (d) show the seasonal cycle of the ice water path (IWP) due to moist and dry cirrus. The IWP is dominated by the ice water at low altitudes (see Fig. 10). Therefore, the seasonal cycle of the IWP reflects the seasonal cycle of cirrus clouds at low altitudes. For moist cirrus, the seasonal patterns of the IWP and the maximum frequency of occurrence located at 14 km (see Fig. 1b) are similar. This indicates that moist cirrus throughout the troposphere are coupled to each other and to convection. On the other hand, by comparing Figs. 8(b) and (d), we can see that, for dry cirrus, the seasonal pattern of the IWP is different from that of the vertical maximum of the frequency of occurrence (which is located near 16 km in the TTL, see Fig. 1b). While high-altitude dry cirrus are mostly located in the deep tropics, low-altitude dry cirrus occur over all latitudes of the tropics, from the equatorial region to the northern and southern edges of the tropics. These behaviors suggest that dry cirrus at low altitudes are decoupled from dry cirrus at high altitudes. We posit that low-altitude dry cirrus are driven by gravity waves' temperature perturbations in the troposphere. Gravity wave activities are expected to be more prevalent in the winter months than the summer months in each hemisphere, consistently with the seasonal pattern of the IWP of dry cirrus clouds in Fig. 8(d).

**5  Summary**

Based on the monthly anomalies of the SPH and temperature in cloudy air relative to all-sky condition, we have separated the population of tropical cirrus clouds detected by CALIPSO into moist cirrus and dry cirrus. We define moist cirrus as those occurring in air that contains more moisture than usual, while dry cirrus occur in air that is colder and contains less moisture than usual.

Moist cirrus are on average located at lower altitudes than dry cirrus. The level of maximum moist cirrus occurrence is near 14 km, i.e., the bottom of the TTL. In comparison, dry cirrus obtain their maximum frequency of occurrence at 16 km. The ratio of the number of moist cirrus to the number of dry cirrus is on the order of 2:1 over all altitudes of the tropical atmosphere, 3:1

below 14 km, 1:1 above 14 km, and 1:3 above the CPT. The majority of dry cirrus are located above 14 km, but there are also dry cirrus below 14 km. Dry cirrus at high altitudes occur in the deep tropics, while dry cirrus at low altitudes occur all over the tropics, from the equatorial region to the northern and southern edges of the tropics.

The seasonal cycle of moist cirrus is consistent with that of tropical convection, while the seasonal cycle of dry cirrus above 14 km is consistent with that of the CPT. There are two maxima in the frequency of occurrence of moist cirrus, one at around 11 °S–12 °S in the austral summer, and the other at around 11 °N–12 °N in the boreal summer. In contrast, dry cirrus above 14 km occur most frequently in the deep tropics in the boreal winter when the CPT is coldest. Dry cirrus below 14 km occur most frequently in the winter months of each hemisphere whence wave activities are strongest. These results suggest that the monthly occurrence of moist cirrus, high-altitude dry cirrus, and low-altitude dry cirrus are driven by different process mechanisms: (i) the occurrence of moist cirrus is driven by the moistening effect of convection, (ii) the occurrence of high-altitude dry cirrus is driven by the adiabatic cooling associated with the BDC as well as wave activities in the TTL and lower stratosphere, and (iii) the occurrence of low-altitude dry cirrus is driven by wave activities in the troposphere.

The IWC in both moist and dry cirrus increases with increasing temperature (decreasing altitude). Thus, dry cirrus—which on average occur at lower temperatures (higher altitudes)—tend to have lower IWCs than moist cirrus. However, at a given altitude, the IWCs in moist and dry cirrus are comparable to one another. Fresh outflow convective anvil cirrus may have much larger IWCs, but subsequent processes during their life cycles such as ice sedimentation and sublimation, and cloud horizontal spreading can decrease the IWCs by several orders of magnitude as shown in previous studies [*Boehm et* al., 1999; *Luo and Rossow*, 2004; *Dinh et al.*, 2010, 2012; *Gehlot and Quaas*, 2012; *Dinh et* al., 2014; *Jensen et* al., 2018; *Gasparini et* al., 2021].

The method proposed here to study cirrus clouds can be applied in model development to improve the representation of cirrus clouds in numerical simulations. We have demonstrated that the spatiotemporal distribution of cirrus clouds is governed by the SPH, temperature, and their variations. Therefore, models would need to accurately represent the SPH, temperature, and their variances in order to accurately simulate the distribution of cirrus clouds. It would be useful to compare between observations and numerical simulations in terms of the frequency and magnitude of the moisture and temperature anomalies and how they affect the occurrence of cirrus clouds. Such a comparison would reveal the specific strategies on how to adjust the model parameterization schemes (e.g., the convection scheme, the gravity wave drag scheme, and/or the microphysics scheme) to improve the representation of cirrus clouds in models.

*Author contributions.* Qin Huang carried out the analysis of the data. Both Qin Huang and Tra Dinh contributed to writing the manuscript.

*Competing interests.* No competing interests are present.

*Acknowledgements.* We thank D. Winker for the discussions at the AGU Fall Meeting in 2019, which motivate this work. We thank Blaž Gasparini and an anonymous reviewer for their feedback and comments which help improving the manuscript.

–